# Open-source personal pipetting robots with live-cell incubation and microscopy compatibility

Philip Dettinger[1,2✉], Tobias Kull[1], Geethika Arekatla[1], Nouraiz Ahmed[1], Yang Zhang[1], Florin Schneiter[1], Arne Wehling [1], Daniel Schirmacher [1], Shunsuke Kawamura[1], Dirk Loeffler [1] & Timm Schroeder [1✉]

Liquid handling robots have the potential to automate many procedures in life sciences. However, they are not in widespread use in academic settings, where funding, space and maintenance specialists are usually limiting. In addition, current robots require lengthy programming by specialists and are incompatible with most academic laboratories with constantly changing small-scale projects. Here, we present the Pipetting Helper Imaging Lid (PHIL), an inexpensive, small, open-source personal liquid handling robot. It is designed for inexperienced users, with self-production from cheap commercial and 3D-printable components and custom control software. PHIL successfully automates pipetting (incl. aspiration) for e.g. tissue immunostainings and stimulations of live stem and progenitor cells during time-lapse microscopy using 3D printed peristaltic pumps. PHIL is cheap enough to put a personal pipetting robot within the reach of most labs and enables users without programming skills to easily automate a large range of experiments.

[1] Department of Biosystems Science and Engineering, ETH Zurich, Basel, Switzerland. [2] Present address: University of Basel, Hebelstrasse 20, 4031 Basel, Switzerland. ✉email: philip.dettinger@unibas.ch; timm.schroeder@bsse.ethz.ch

Liquid handling and pipetting tools are important in biological research but are often limited to specific industries and tasks. They automate time consuming and repetitive experimental procedures, reduce personnel cost, and can enable otherwise impossible complex dynamic or high-throughput experiments. Many robotic liquid handling platforms exist, but they are useful only for a limited range of applications, usually in industrial high-throughput settings[1,2]. First, liquid handling machines are very expensive and generally require a lot of room[3]. Second, programming these robots for new projects is very complex and slow, usually requiring dedicated experts. This makes them impractical for most academic groups with many small-scale projects with constantly changing low throughput requirements. While costs for purchasing or building pipetting robots are decreasing[3–10], researchers remain limited by programming requirements and the experimental procedures that liquid handling robots can automate.

This is true in particular for live cell cultures during time-lapse microscopy. Manually pipetting complex and dynamic liquid sequences during live cell microscopy experiments is labour intensive, requiring precisely controlled interventions by skilled researchers whose time is blocked during these long procedures. This also often interrupts culture conditions and can obscure fast responses in the observed cells. Automated pipetting would thus be very helpful for these experiments but is particularly challenging for existing robots. Complex and expensive machinery is typically used to move culture vessels to/from dedicated microscopes, which also prevents rapid culture media changes during live cell microscopy. Liquid handling robots that are integrated with advanced microscopes have also been developed[8]. However, most of these systems lack environmental control equipment and are limited to specific imaging modalities that are not suitable for observing all cellular processes[8].

Other automated methods utilizing pump arrays connected to a single culture vessel have been developed that can conduct pre-programmed sample washing procedures during microscopy[11]. While these methods are optimized for small numbers of culture conditions, they do not scale well to studies with many different conditions to be simultaneously analysed, as each vessel generally requires a complete set of pumps, fluids, and microscopes. This also demands significant technical expertise to implement in large numbers and can become cost- and time-prohibitive very quickly. Other methods that utilize single pipets to transfer liquids from multi-well plates to microfluidic devices during microscopy experiments have been published[12], but these approaches are limited to specific microfluidic devices and are not optimized for diverse and simultaneous test conditions.

Large-scale integrated microfluidics[13–15] enable automated cell culture interventions over long periods during time-lapse microscopy under environmentally controlled conditions[14,16]. However, available designs suffer from several shortcomings that interfere with their widespread implementation and use[17]. They require significant investment in training and equipment for production, as well as extensive experience for successful operation[17]. Given the porous nature of the polydimethylsiloxane (PDMS) used to construct them, they cannot be reused, further increasing the time and cost required for their use while reducing their reproducibility between users[18,19]. Highly specialized chips are generally optimized for one specific procedure, further limiting their use in academic lab environments with constantly changing experimental requirements[20]. Due to a lack of standardization between designs, microfluidic chips are often not compatible with commonly available scientific equipment for single-cell analysis without time-consuming and complex cell isolation procedures[21]. Microfluidic devices intended for efficient long-term time-lapse microscopy and endpoint analysis have been developed[22–26]; however, no single device has been proposed that can replicate the ease of use, cost efficiency, and interoperability of standard cell culture vessels such as multi-well plates.

Here, we present a liquid handling robot that can automate a variety of experimental procedures in standard cell culture vessels. The pipetting helper imaging lid (PHIL) is small enough to be installed almost anywhere, including on most microscope stages, and automates complex liquid handling sequences in standard well plates while also allowing conditioning of the incubation gas atmosphere. PHIL is cheap to self-manufacture by inexperienced users using common 3D printing equipment and fast assembly with commercial parts. In combination with a custom made intuitive graphic user interface (GUI), PHIL enables a variety of experimental procedures including, for example, automated immunostaining of fixed cells and dynamic interventions in live cell cultures during long-term time-lapse microscopy experiments.

## Results

**Design of a stage top robot for automated cell culture interventions.** We developed a liquid handling robot that utilizes a 5-bar planar geometry to position between 1 and 10 reusable pipets in any region of a standard multi-well plate (Fig. 1a). It is small and light enough to install on microscope stages (Fig. 1a). Cell culture plates therefore do not have to be removed from the microscope for pipetting. All components are 3D printable or cheap commercially available standard parts (Fig. 1a, b).

Pipet positions are determined by two stepper motors that rotate independent 65 mm arms connected in a planar configuration by two freely rotating 145 mm links (Supplementary Fig. 1 and Supplementary Movie 1). One of these links has a claw protruding 1 cm perpendicular to its axis of rotation (Supplementary Fig. 1). These lengths were chosen to optimize the effective working area while keeping the device small enough for stage-top use, and to eliminate the possibility of joint configurations that could render the device inoperable. All freely rotating connections feature an 8 mm steel ball bearing to prevent binding. Robot arms are raised and lowered up to 45 mm by two stepper motors placed on each side of the manipulator arms and connected to the arm carrying platform via a 3D printed screw drive mechanism (Supplementary Fig. 2). Limit switches installed at the base of each arm and parallel to the screw drives provide homing functionality during operation. The 5-bar planar configuration was selected over Cartesian designs due to chamber volume and speed requirements.

PHIL can be configured to dispense or aspirate liquids through 1–10 separate tube channels and easily produced reusable pipets. The 23 gauge stainless steel tubes, utilized as pipets, are tightly clustered in a mounting bracket held by the robotic arm. Each pipet is connected to a custom-designed peristaltic pump and media reservoir via 1.07 mm diameter Teflon tubing (Fig. 1c). Control of flow rates through pipets is determined by the rotation of 2 stepper motors connected to rotors in up to 10 custom-designed 3D printed pumps, each capable of 2–100 μL/s flow rates (Fig. 1c). Peristaltic pumps were chosen as the ideal pumping method because they do not require reconfiguration for distinct media reservoir volumes and enable fast possible pump speeds.

For programming and operating PHIL, we programmed a custom GUI. In contrast to the very complex programming skills required for most existing liquid handling robots, our GUI has been designed for easy intuitive programming by inexperienced users for precise and automated handling of any target multi-well plate design. Briefly, the Matlab GUI is divided into four sections: Calibration, Manual control, Scripted Procedures, and "Remote"

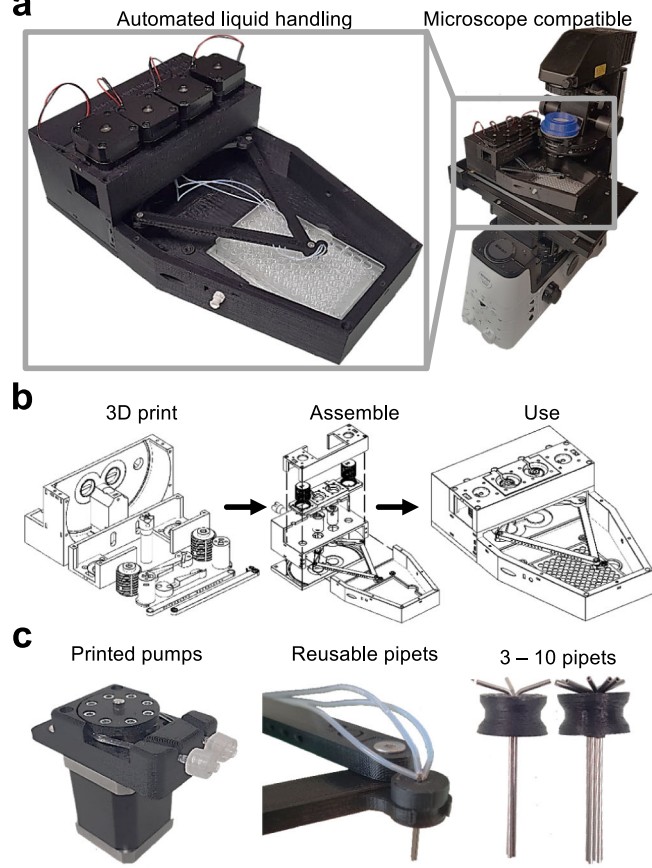

**a** Automated liquid handling    Microscope compatible

**b** 3D print    Assemble    Use

**c** Printed pumps    Reusable pipets    3 – 10 pipets

**Table 1 PHIL pipet movement and liquid flow enables rapid automatic media exchange in common multi-well plate types.**

| X/Y | Z | Pipet |
|---|---|---|
| 5 cm/s | 1.5 cm/s | 2–100 µL/s |
| **Plate type** | **Tme/well (s)** | **Time/plate (min)** |
| 6 well | 40 | 4.1 |
| 24 well | 20 | 8.6 |
| 96 well | 3.5 | 5.8 |
| 384 well | 2 | 12.8 |
| ibidi µ-Slide VI | 4.5 | 2 |
| MicroAmp® PCR | 7 | 8.5 |

**Table 2 PHIL can be rapidly deployed with minimal expenses and expertise.**

| Print time | Weight | Footprint area | Cost | Assembly time |
|---|---|---|---|---|
| 7 days | 950 g | 20 × 30 cm | 600 USD | 2.5–7 h |

All parts can be printed in 1 week and assembled with commercial parts for under 600 USD in 2.5–7 h. Minimal technical expertise is required for assembly and installation.

**Fig. 1 PHIL, the Pipetting Helper Imaging Lid robot with a simple design for complex automated liquid handling procedures. a** PHIL pipets reach areas of common multi-well plate types. The small and light robot also fits microscopy stages and can automate complex live cell culture interventions without obstruct imaging. **b** All parts are commercially available or can be 3D printed using easily available equipment. Assembly is fast and requires no specialized expertise. **c** PHIL utilizes peristaltic pumps connected to reusable pipets. Peristaltic pumps assembled from 3D printed and commercial parts (left) drive flow through 1–10 separate tubes (middle), each ending in separate reusable pipets (right).

mode. Calibration provides automated pump calibration as well as control over stepper motor speeds and settings including motion and pump flow rates. Manual control allows users to move the pipets to any location in the working area, define custom locations, and dispense defined volumes. Scripted procedures allow users to define a series of liquid removal and dispensing commands in wells of various standard devices over time. Remote mode allows PHIL to receive triggered pipetting instructions in real time from other devices. For ease of use and a shallow learning curve, users define a target excel file that the GUI loads and reloads on a defined interval while carrying out any instructions contained within. The GUI and instructions for its operation are available online (https://bsse.ethz.ch/csd/Hardware/PHILrobot.html, https://github.com/CSDGroup/PHIL).

An Arduino Mega connected to a custom shield circuit receives commands from our GUI via USB serial communication and translates them into coordinated stepper motor rotations via an Arduino sketch (Accelstepper Arduino library) for speed and acceleration control. All electronics are housed in a custom enclosure for cable management and easy transport.

The cell culture vessel, robotic arms, and reusable pipets are enclosed inside a sealed incubation chamber for temperature, humidity and gas control, with standard connections for conditioned gas inputs as well as humidity and temperature sensors. It has a 2 mm-thick transparent poly(methyl methacrylate) (PMMA) window above the multi-well plate for light access in inverted microscopy. PHIL's dimensions are ~200 × 300 × 100 mm ($W \times L \times H$); The maximum height for cell culture vessels is 45 mm, optimized for a short microscopy light path length while providing sufficient vertical space for travel of the pipet arm between, e.g. wells of common multi-well plates.

PHIL has a maximum speed of 5/5/1.5 cm/s in $X/Y/Z$ (Table 1). Common multi-well plates require 2–40 s for complete media replacement per well, and 4–13 min for all wells (Table 1).

**Production and assembly of PHIL robots.** All PHIL components were printed on a standard 200 mm by 200 mm build plate in 10 printing sessions with an Ultimaker 2 extended+ 3D printer using default settings (Table 2). Detailed part lists and instructions for non-experienced users are available online (https://bsse.ethz.ch/csd/Hardware/PHILrobot.html, https://github.com/CSDGroup/PHIL). Each printing session required 8–26 h (Table 2). 3D printing was chosen as the ideal fabrication method due to its widespread availability, ease of open-source sharing, local reproducible production and lightweight products. All components were printed in acrylonitrile butadiene styrene (ABS) filament due to warping observed in alternative materials (PLA) when maintained at 37 °C and >95% relative humidity for extended periods. The total weight of the robot body and manipulator is 953 g, compatible with commonly available automated microscope stages. Users who wish to outsource the 3D printing of PHIL components can do so by uploading the respective.stl files to a variety of online services (e.g. shapeways.com, hubs.com, i.materialize.com, etc.).

Importantly, we optimized all used electronic components for easy accessibility for inexperienced users. The off the shelf components for a basic PHIL with 3 pumps and polyethylene terephthalate (PET) tubing cost below 600 USD. A PHIL with 10 pumps and polytetrafluoroethylene (PTFE) tubing and a pre-assembled printed circuit board (PCB) costs up to 800 USD and can be produced with minor soldering steps. Complex wiring and soldering processes were removed using an open-source

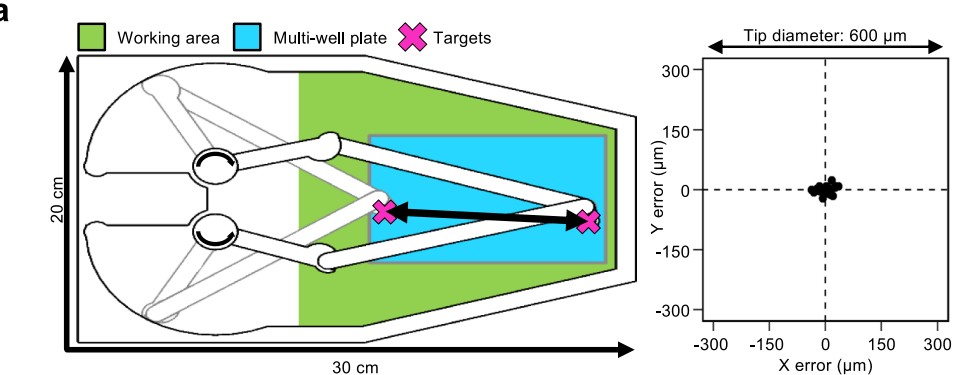

**Fig. 2 PHIL reliably and accurately positions pipet tips in multi-well plates. a** Pipets move to precise locations within full multi-well plate dimensions. Alternatively, this area can hold e.g. 1 × 10 cm dish, 4 × 3 cm dish, 4× ibidi channel slides, 96× PCR tubes, or other vessel smaller than a multi-well plate. When moving between the centers of 2 wells across a 96-well plate (left), pipet locations varied by only ± 50 μm in X and ±30 in Y (right, n = 57 trials).

PCB, which can be ordered online (for ~20 USD, instructions available at https://github.com/CSDGroup/PHIL) or produced in house (https://bsse.ethz.ch/csd/Hardware/PHILrobot.html, https://github.com/CSDGroup/PHIL) and connects directly to an Arduino Mega 2560 Rev3. To confirm the ease of production by inexperienced users, we tested device production with five separate biology Ph.D. students with no engineering experience in our group. Following the provided instructions (Supplementary Notes 1 and 2), they successfully assembled functional devices without circuit boards in 2.5–4 h (Supplementary Movie 2) without further assistance. Circuit boards were either self-fabricated in an additional 3 h, or prefabricated circuit boards were commercially acquired for ~20 USD (instructions available at https://github.com/CSDGroup/PHIL).

**Robust media exchange in multi-well plates via automated pipetting.** Several rounds of optimization lead to the current design as the optimal compromise between accuracy and applicability. To test the reliability and accuracy of PHIL movements, we positioned one of the pipets directly above the center of well F12 in a 96-well plate. We then moved the pipet across the whole plate to well E1 and back (240 mm travel distance) and imaged its position upon its return (Fig. 2a). Over 25 repetitions, the maximum positioning error of the pipet was 50/30 μm in X/Y, which is <1/100th of a 96-well plate well diameter, and thus negligible (Fig. 2a). These error rates could be reduced even more through the use of more precisely machined metal components and servomotors, but these unnecessary improvements would increase cost and weight, thereby reducing stage-top compatibility.

Several rounds of testing identified 1/16th microstepping as the optimal balance of speed and torque for PHIL media pumps. For their calibration, we instructed each pump stepper motor to complete 100,000 1/16th microsteps, or approximately 31 full rotations, and weighed the pumped water volumes to calculate the μL volume per step, which was then used in the GUI. Based on the dimensions of our peristaltic pump, the theoretical limit of resolution is 0.4 μL/step.

We tested PHILs' pump accuracy by pumping 10, 25, 100, and 1000 μL of water and weighed the pumped volumes (10 repetitions). 10, 25, 100, and 1000 μL target volumes resulted in 7–11, 23–27, 98–101, and 998–1002 μL samples, respectively (Fig. 3a), similar to manual pipetting and other robots[7,10,27–30]. The resulting volumes varied from the target volume by ± 2 μL regardless of the target volume, suggesting droplet retention at the pipet tip rather than pump inaccuracies as the source of this variation (see below). When testing the impact of pumping speed

on accuracy, resulting volumes were within 1–2% of the target volume for 10, 25, and 50 μL/s. Pumping accuracy dropped to 95% at 100 μL/s (Fig. 3a), which could be compensated for by automatic recalibration if higher accuracy is required at faster pumping speeds. The current pump design is optimized for accuracy over speed and 100 μL/s was used as a maximum pumping rate to prevent infrequent stalling at higher flow rates. For higher flow rates, more powerful motors could easily be implemented with minor calibrations.

To further test the pipetting accuracy when mixing different input channels, we loaded one pipet with 100 μM FITC in phosphate buffered saline (PBS) and another with PBS. PHIL was then instructed to mix these to final concentrations of 0–100 μM FITC in 96-well plates. These dilutions were also manually pipetted in the same plates, and the resulting FITC fluorescence was quantified via fluorescence microscopy (Fig. 3a). Mean FITC fluorescence in each condition was comparable between manually and automatically produced conditions.

Dynamic changes of cell culture media require accurate media aspiration and addition. To test PHILs' capacity to change media composition in standard cell culture vessels, we programmed PHIL to position pipets 0.5 mm above 96-well plate bottoms before aspirating 100 μl PBS and adding 100 μl 100 μM FITC at 50 μl/s and imaged the resulting fluorescence. The fluorescence quantification confirmed the expected 20 s single pulse of FITC fluorescence (Fig. 3b and Supplementary Movie 3). This was also possible in rapid succession when programming a 120 s sequence of 15 s oscillations without pauses between liquid transitions (Fig. 3c and Supplementary Movie 4). Deeper wells could result in inefficient media exchange, which could be compensated for by utilizing longer pipets (Supplementary Fig. 3).

2 h FITC:PBS oscillations over 48 h confirmed that PHIL reliably maintains dynamic culture conditions over long periods (Fig. 3d and Supplementary Movie 5). Arbitrary pipetting programs can be applied simultaneously to different wells (Fig. 3e).

**Eliminating contamination during automated pipetting with reusable pipets.** We chose a design with reusable pipets and instead of single-use pipets to minimize the incubation chamber height, maintain PHILs compatibility with various microscopes, and reduce design complexity. While the use of reusable pipets can cause contamination and volume changes, these are unproblematic for many applications, and can be efficiently prevented by washing procedures. To quantify carry-over, PHIL pipets were moved from wells containing FITC solution to PBS wells and the volume of transferred FITC contaminant was measured by

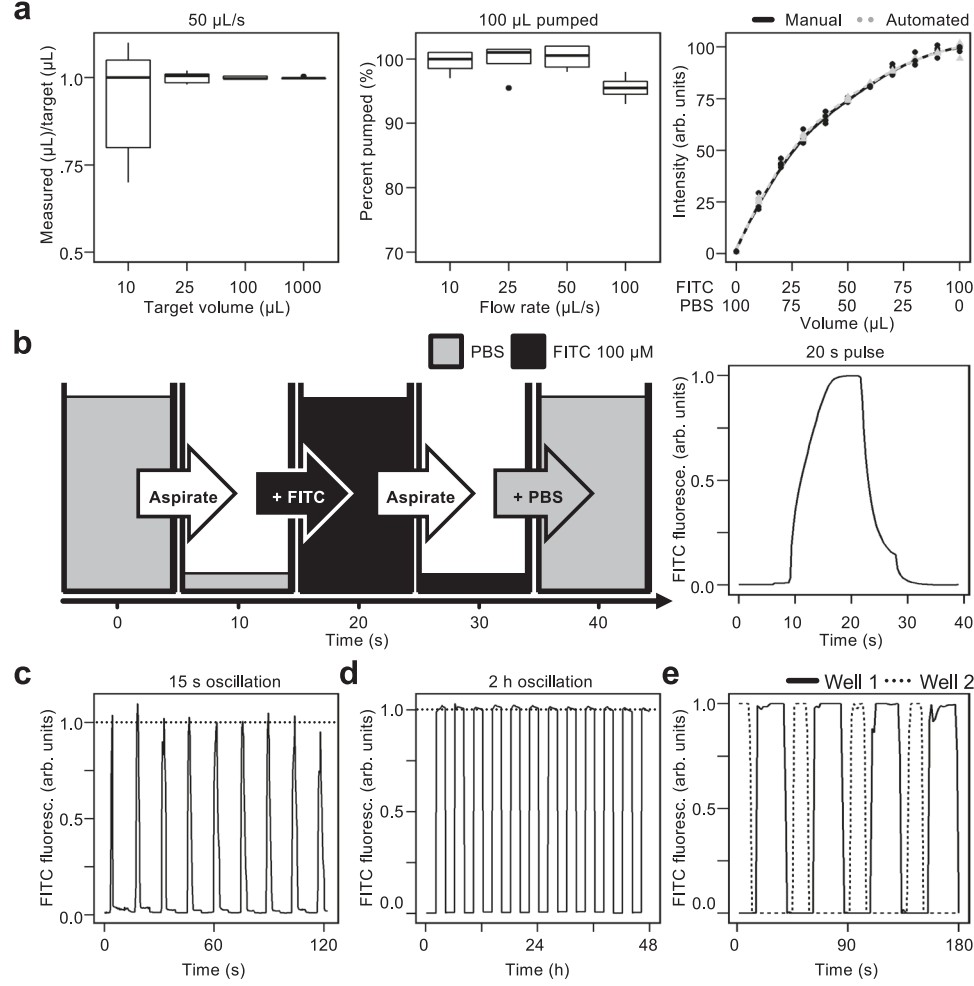

**Fig. 3 PHIL accurately conducts pipetting procedures in standard culture vessels. a** PHIL accurately pumps precise volumes. (Left, $n = 25$ samples) 10, 25, 100, and 1000 µL of pumped $H_2O$ varied from the target volume by ±1–2 µL. (Middle, $n = 16$ samples) 100 µL $H_2O$ pumped at 10, 25 and 50 µL/s varied from the target volume by ±1–2%. When pumped at 100 µL/s the target volume was 95 ± 2 µl. (Right, $n = 88$ samples) Manual and automated dilutions of 100 µM FITC and PBS are comparable. Box plot features, from top to bottom, represent maximum, 3rd quartile, median, 1st quartile, and minimum values. **b** Quantification of a FITC dextran pulse over 20 s. Pipets aspirate PBS (grey), add 100 µM FITC (black), aspirate FITC, and add PBS. Mean FITC fluorescence imaged every 0.5 s reached peak fluorescence in 5 s when pumped at 50 µL/s. **c** Rapid 15 s oscillating FITC patterns created by automated pipetting in a 96-well plate. Mean FITC fluorescence imaged every 0.5 s reached ± 5% of maximum fluorescence (dashed line) on a 15 s interval. **d** Consistent 2 h FITC oscillations maintained over days. Mean FITC fluorescence imaged every 5 min reached ± 1% of maximum fluorescence (dashed line) on a 2 h interval for 2 days. **e** Distinct FITC oscillations can be maintained simultaneously in different wells. Mean FITC fluorescence in 2 wells imaged every 1.5 s reached ± 1% of maximum fluorescence during 20 s (dashed line) and 30 s (solid line) pulses.

fluorescence microscopy (Fig. 4a). The measured carry-over was approximately 1 µl. To reduce this contamination, we implemented an automated washing step in which pipets are washed with target media in an unused well before entering the target well (Fig. 4a). The resulting reductions in cross contamination for 1, 4, and 8 automated wash steps (requiring 3.5, 14, and 28 s, respectively) were 1:1000, 1:100,000, and >1:100,000, respectively (Fig. 4a). These contamination values are compatible with most typical experiments including immunostaining procedures, toxicology studies, and even for sensitive readouts like cytokine stimulations of live cell cultures.

PHIL can be simultaneously equipped with 3–10 pipets with their own separate tubing. Contamination between wells increased with pipet count with the required washing steps doubling between 3 and 10 pipets (Supplementary Fig. 4).

**Automated immunostaining pipetting with PHIL.** Immunostaining procedures require numerous liquid exchange steps binding personnel over long periods of time and thus are ideal for automation. Having confirmed pipetting accuracy and reproducibility, we tested PHIL's ability to automate time consuming immunostainings of decalcified murine bone marrow and cerebrum. Fixed samples were subjected to automated PHIL immunostaining, and imaged by confocal microscopy (Fig. 5a). The resulting images showed immunostained subcellular features and intact structures (Fig. 5a) comparable to the result of manual staining (Supplementary Fig. 5), demonstrating successful and gentle pipetting by PHIL.

PHIL immunostaining also worked well on flow sensitive fixed single non-adherent hematopoietic progenitor cells. We sorted 3 different subsets of primary murine myeloid progenitor cell types based on GATA2VENUS fusion protein reporter expression (Supplementary Fig. 6)[31], seeded freshly sorted cells in 96-well and fixed with formalin. The automated immunostaining procedure involved 10 different incubation and washing steps lasting 12 h from four different types of staining and washing input solutions (Fig. 5b; see the "Methods" section). One sample was not incubated with primary antibody, serving as a control for

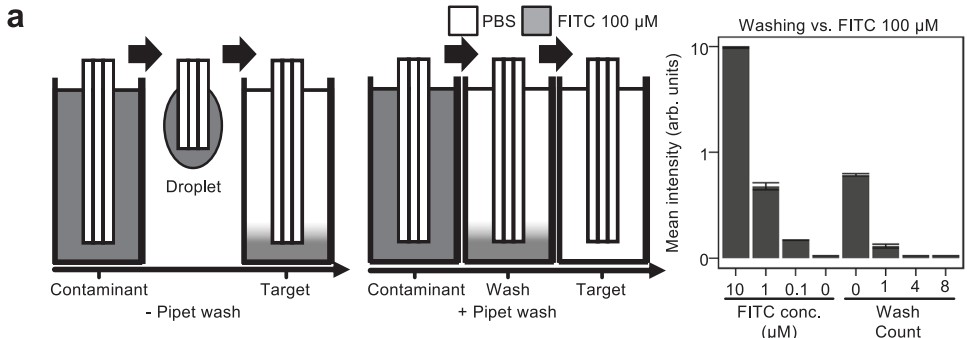

**Fig. 4 Washing pipets between samples eliminates contamination. a** Cross contamination is eliminated through automated washing procedures. (Left) Pipets carry sample droplets between wells. (Middle) Pipet washing in target medium in unused well reduces contamination. (Right) Pipet washing reduces carry-over contamination. A FITC dilution curve (left) allows inference of FITC concentration from fluorescence brightness. 0, 1, 4, and 8 wash steps (right) reduce contaminant concentration in the target well to 1%, 0.1%, 0.001%, and 0.0001% of the contaminant source well concentration (100 µM, $n = 24$ samples). Box plot features, from top to bottom, represent maximum, 3rd quartile, median, 1st quartile, and minimum values.

cross contamination. After automated immunostaining, the samples were imaged, and single-cell fluorescence intensities were quantified (Fig. 5b). The resulting fluorescence intensity distributions of VENUS-negative, VENUS-medium and VENUS-high cells were the same between automated immunostaining and the initial FACS quantification, demonstrating the reliability and reproducibility of automated PHIL immunostaining. No signal was detected in the secondary only control, confirming the absence of cross-contamination by automated pipetting.

To quantify the ability of PHIL to handle flow-sensitive samples that require very gentle pipetting to not get washed away or displaced upon liquid exchange, we loaded anti-CD43-biotin coated[32] ibidi µ-Slide VI 0.4 channel slides with freshly sorted human cord blood CD34 + HSPCs and placed them in culture for 24 h, then fixed with 4% formalin solution and washed once with PBS manually and at varying automated flow speeds (Fig. 5c). Complete aspiration of formalin solution and one time dispensing of 100 µl PBS were conducted in the center of media reservoirs on opposing ends of the channels in order to induce unidirectional flow. Liquid aspiration and dispensing were conducted at the same speed. Cells were imaged via fluorescence microscopy before and after washing, and the resulting change in cell counts was recorded via automated segmentation[33]. Manual washing by an experienced researcher was estimated to have occurred at 10 µL/s based on observed wash volume and time (e.g., 100 µL over 10 s). Manual washing resulted in an average cell loss of 12.6%, while automated washing significantly improved cell retention with 2 µL/s flow rates only changing cell counts by 3.6% (Fig. 5c). Further reductions in flow rate are possible with PHIL, but increase processing times beyond what is practical for our purposes.

**Automated pipetting of long-term live cell cultures**. To test PHIL's compatibility with live cell cultures, we compared cell proliferation and viability of murine embryonic stem cells (ESC) over 36 h in 96-well plates when cultured on a microscope in PHIL versus our standard live cell microscopy incubator. The mean time to first division for ESCs cultured in our standard culture chamber and in PHIL were 12.3 and 12.5 h, respectively, while mean cell counts per input cell after 36 h were 2.6 and 2.3, respectively (Fig. 6a). Thus, incubation in PHIL did not alter ESC survival or proliferation.

To test PHIL pipetting in live cell cultures requiring very gentle pipetting, freshly sorted non-adherent primary murine granulocyte-monocyte progenitors (GMP) were imaged with and without automated media exchange. The resulting time-lapse image sequences were manually tracked to quantify cell displacement. Unwashed versus washed cells moved 2.1 versus 1.8 µm on average (Fig. 6b), washing therefore did not displace cells. We observed no cell loss in either condition (Supplementary Movie 6) and successfully maintained all cell identities through the entire process, confirming the utility of PHIL even in live cell experiments requiring very gentle pipetting.

The quantification of weak fluorescent signals in individual tracked cells after dynamic cytokine stimulation requires extremely gentle pipetting to avoid cell displacement and the resulting errors in nuclear segmentation and tracking[22,23,34–41]. So far, demanding custom-designed microfluidic chips were necessary to enable the required automated cytokine stimulation by diffusion[22,23,42]. We therefore tested PHIL pipetting of TNFα pulses for subsequent quantification of p65 signalling activity dynamics in individual non-adherent mouse GMPs. We used GFP-p65/H2B-mCHERRY murine GMPs[43,44], where H2B-mCHERRY provides a fluorescent nuclear label (chromatin) for nucleus segmentation. GFP-p65 resides in the cytoplasm versus nucleus in its inactive versus active state, thus allowing the quantification of p65 activity dynamics (Fig. 7a, b). We plated freshly sorted GMPs (Supplementary Fig. 7) and incubated them on the microscope stage during automated media exchange and time lapse imaging. GMPs were imaged every 9 min using confocal microscopy and individual GMPs were tracked for quantification (Fig. 7c)[35]. All cells were imaged without TNFα for 1 h to establish baseline p65 activity. Media in different wells were then changed to 40 ng/mL TNFα with refreshment every 90 minutes, or to 0 or 40 ng/mL TNFα for 15 min and then back to 0 ng/mL TNFα (Fig. 7d). GMPs without TNFα stimulation showed 89% cells without p65 activation (non-responders), while the various TNFα stimulation dynamics activated p65 in 76–86% cells as expected. The response rate in the unstimulated condition is comparable to previously observed manually pipetted experiments Kull et al.[45] and demonstrates PHIL's capacity to conduct complex dynamic live cell stimulations without cross contamination (Supplementary Fig. 8). TNFα has a very short half-life of 15–30 min[46]. Refreshing TNFα over time could therefore have an effect on the p65 activity dynamics. Indeed, a single TNFα pulse led to less sustained responders than TNFα stimulation with refreshment every 90 min (Fig. 7d).

**Discussion**

We describe the design and performance test of an open-source automated compact lightweight and cheap liquid handling robot that can be quickly deployed with minimal investment in time and resources. Using 3D printed parts allow reduced size and

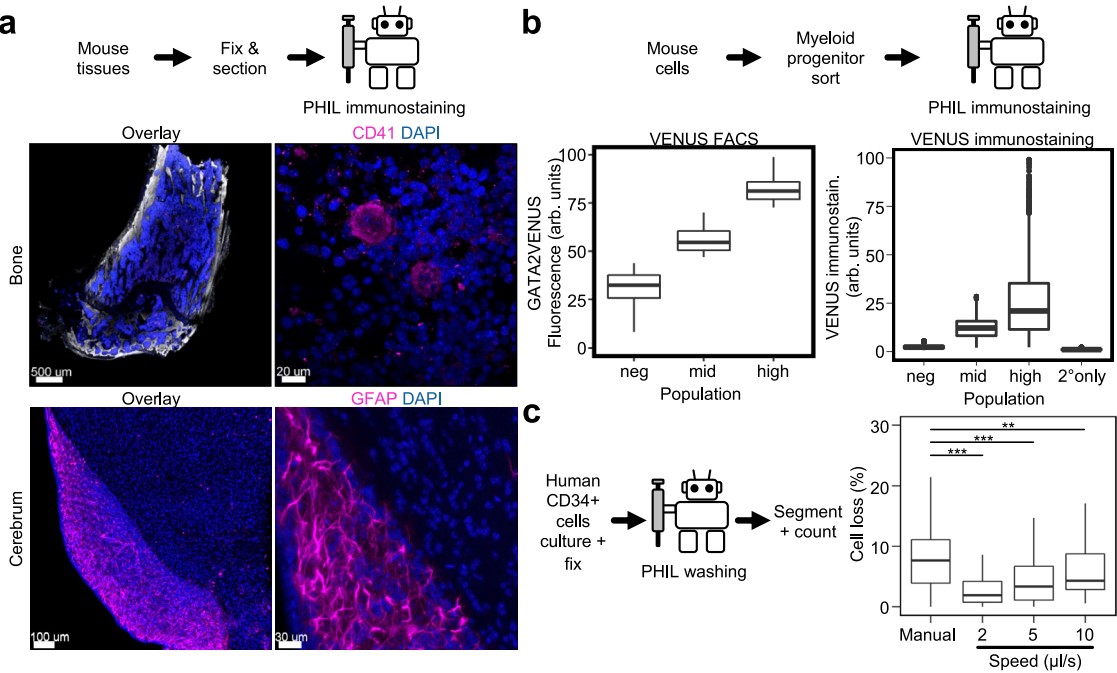

**Fig. 5 PHIL successfully automates single cell and whole tissue immunostaining procedures. a** Fixed and cleared murine bone and cerebrum samples were sectioned before automated PHIL immunostaining and imaging. PHIL immunostaining preserved normal tissue morphology with intact macroscopic (left) and subcellular (right) features. Compare Supplementary Fig. 5 for comparison to a manually pipetted sample ($n = 2$ experiments). **b** Murine myeloid progenitors were sorted into GATA2VENUS-negative, -middle, and -high populations, plated and fixed with 4% formalin, and PHIL immunostained against VENUS with and without primary antibodies and imaged. VENUS quantification of PHIL immunostained samples showed Venus negative, mid and high in both microscopy and FACS analyses ($n = 2$ experiments) and showed no antibody cross-contamination. The relative brightness of the different populations followed the same trend when observed via FACS or microscopy and was consistent with previously conducted manual immunostainings on the same populations[31]. Box plot features, from top to bottom, represent maximum, 3rd quartile, median, 1st quartile, and minimum values. **c** Automated washing improves cell loss for flow sensitive samples over manual pipetting. Human umbilical cord CD34 + hematopoietic stem and progenitor cells, cultured for 1 day in a μ-Slide VI[0.4] prior to fixation, were washed once with 100 μL manually or with 2, 5, and 10 μL/s flow. Culture channels were imaged and counted before and after washing. Mean cell loss for manual versus 2, 5, and 10 μL/s were 12.6 versus 3.6%, 4.3%, and 6.1% respectively (p-values from top to bottom: 0.0094, 0.0009, 0.0004, two-tailed $t$-tests with Benjamin–Hochberg multiple test correction, $n = 2$ experiments). Box plot features, from top to bottom, represent maximum, 3rd quartile, median, 1st quartile, and minimum values.

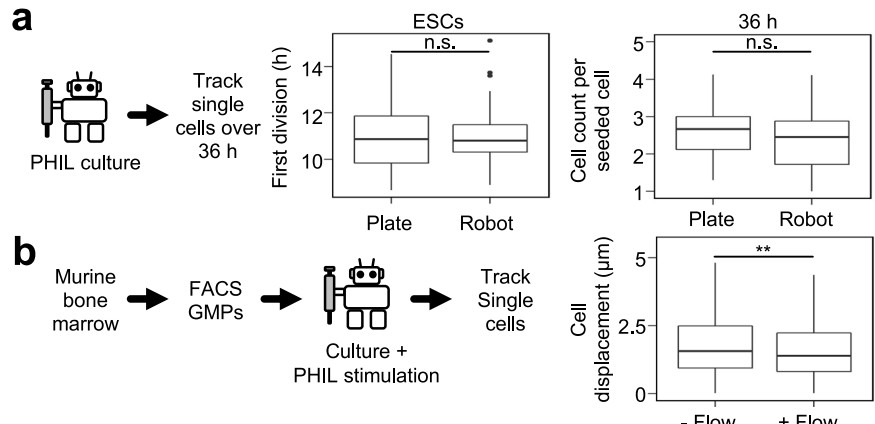

**Fig. 6 PHIL successfully conducts media exchanges without disrupting sensitive samples and cell cultures. a** Mouse embryonic stem cells (ESCs) proliferate normally during automated PHIL culture experiments. ESCs were cultured and imaged in PHIL for 36 h. Mean time to first division with and without PHIL were 12.3 and 12.5 h, respectively ($p = 0.72$, two-tailed $t$-test, $n = 3$ experiments). ESC counts per seeded cell after 36 h of culture with and without PHIL were 2.6 and 2.3 cells, respectively ($p = 0.32$, two-tailed $t$-test, $n = 3$ experiments). Box plot features, from top to bottom, represent maximum, 3rd quartile, median, 1st quartile, and minimum values. **b** PHIL media exchanges do not interrupt cell tracking procedures. Murine GMPs cultured on in CD43-coated 24-well plates wells were imaged and tracked during automated media exchanges. Mean GMP movement or displacement in unwashed versus washed wells were 2.1 versus 1.8 μm, respectively, did not displace cells ($p = 0.003$, two-tailed $t$-test, $n = 3$ experiments). Box plot features, from top to bottom, represent maximum, 3rd quartile, median, 1st quartile, and minimum values.

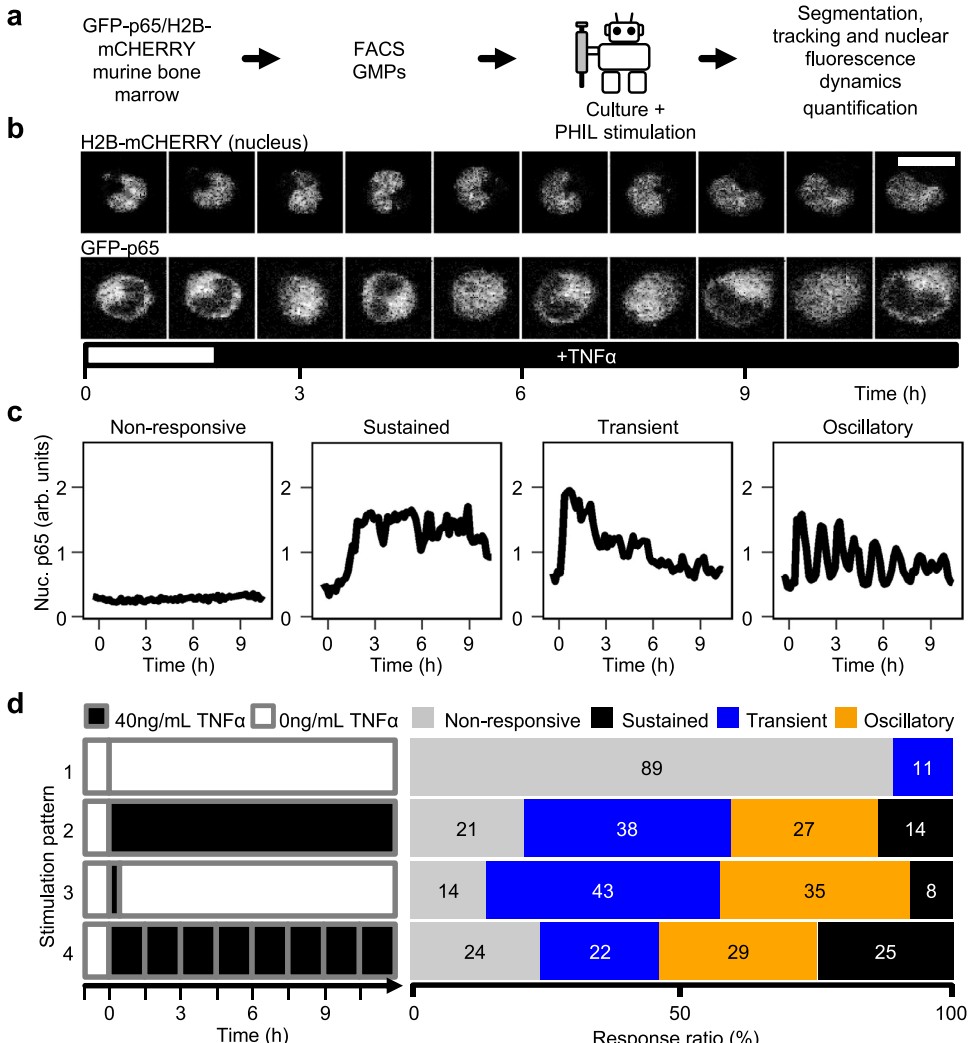

**Fig. 7 PHIL automates parallel cytokine stimulation in sensitive cell cultures without media cross contamination or disrupting time-lapse imaging.**
**a** Freshly sorted GFP-p65/H2B-mCHERRY GMPs were imaged during PHIL culture and stimulation before cell segmentation and tracking (scale bar: 10 μm).
**b** TNFα stimulation results in the rapid translocation of GFP-p65 into the nucleus. GFP-p65 is imported (active) and exported (inactive) from the nucleus in response to a single TNFα stimulation ($n = 3$ experiments). **c** Different observed p65 activity dynamics in response to TNFα stimulation. Example traces of non-responsive, sustained, transient, and oscillatory dynamics. **d** Automated PHIL live cell exposure to different temporal TNFα stimulation patterns during time-lapse imaging experiments. GFP-p65 location was imaged for 1 h prior to stimulation with TNFα. TNFα at a final concentration of 40 ng/mL was then added to TNFα stimulated wells without further medium change (pattern 2), removal of TNFα after 15 min (pattern 3), or TNFα refreshment every 90 min. Nuclear p65 response dynamics vary with the applied TNFα stimulation pattern. 89% of GMPs without TNFα stimulation (pattern 1, $n = 91$ cells) showed no p65 activation, comparable to manual pipetting (Supplementary Fig. 8), showing the absence of cross-contamination. In contrast, between 76% and 86% of GMP exposed to TNFα (patterns 2–4) responded to the stimulus with varied response compositions. More GMPs (25%) showed a sustained response with TNFα refreshed every 90 min (pattern 4, $n = 62$ cells) than with TNFα added once without (pattern 2, 14%, $n = 70$ cells) or with washing it out after 15 min (pattern 3, 8%, $n = 88$ cells), showing the effects of successfully repeated TNFα exposure ($n = 3$ experiments).

weight while keeping its cost way below commercially available machines[2,3,10,29,47,48]. In our hands PHIL could be rapidly constructed by inexperienced users without prior training. This can further be simplified by outsourcing the circuit board assembly to affordable manufacturers (e.g. pcbway.com, jlpcb.com, etc.) by simply uploading the provided circuit production files. Several low-cost open source devices have previously been published that still require significant amounts of programming experience and are not generally compatible with stage top microscopy or other frequently used experimental techniques[3,5]. Here, we combine low cost, rapid deployment, and an intuitive custom open-source GUI into a single device that can be operated in a variety of lab environments. PHIL has a very compact footprint requiring less

dedicated benchtop space than almost any other automated liquid handling robots[1,2,6]. Its robust design enables installation in temperature-controlled enclosures such as refrigerators and incubators and on microscope stages. Several motion control configurations have been utilized for automated pipetting, most frequently Cartesian systems[3,5] or selective compliance articulated robot arms (SCARA)[7,49], but none are as compact as the 5-bar planar arm configuration used in PHIL. It reduces the required interior volume of the device enabling a reduced incubation volume and a more effective isolation of samples from ambient conditions which is crucial for demanding live cell cultures. We found this configuration to also increase the travel speeds between target locations.

PHILs ability to reliably and accurately manipulate the contents of common culture vessels on extremely short time intervals, over long periods, and in multiple parallel vessels is comparable to commercial devices[1,2]. Pipetting procedures automated by PHIL are not limited to the simple paradigm utilized here in which the pipet tip is always placed at the well bottom. With minor programming changes, PHIL could also conduct a variety of improved pipetting procedures such as low turbulence pipetting, where the pipet tip is continuously lowered during aspiration and raised during addition so that it is always just below the liquid surface. To simplify pipetting procedures and reduce the risk of cross contamination between samples, PHIL utilizes dedicated pipets for each of the up to 9 dispensed liquid channels instead of a single pipet connected to multiple pumps. When a single pipet is utilized for serial liquid dispensing, dead volumes present in pipet connections can lead to the incomplete removal of one liquid prior to the addition of another. This issue would have to be prevented by large volume tube washing with the target fluid prior to dispensing. However, many experimental procedures utilize very expensive reagents and keeping used volumes as low as possible is crucial. In addition, the risk of sample contamination associated with a single pipet configuration would limit the experimental complexity PHIL can achieve. Due to the modular design of the pipet mounting bracket and simplicity of our GUI, researchers can easily modify the provided 3D models to choose between configurations with 1–10 separated pipets and liquid channels.

Using a design without removable pipet tips can result in contamination from liquid droplets clinging to the tips of the pipets as they move from well to well. The volume of these droplets increases with increasing numbers of pipets in the bundle. The number of pipets should therefore be reduced to the minimal number required for different experiments. Cross contamination is efficiently reduced by user defined pipet washing procedures between pipetting steps. The remaining extremely low carry over between wells is irrelevant for most experiments, but must be quantified and controlled for sensitive analytical experiments.

Our device is highly scalable and versatile. PHIL's 2.5–7 h assembly time, 600 USD base price, and 20 × 30 cm footprint mean that inexperienced users can rapidly deploy multiple devices for streamlined sample processing workflows or dedicated workstations. The modular design which allows the robot to be printed on any 3D printer and easily assembled also allows for novel geometries and functionalities to be added with minimal effort. Altering characteristics like pump count, chamber height, and arm geometries can be accomplished with simple 3D modelling software. Our 3D printed peristaltic pumps can accept any liquid reservoir and allow users to rapidly change their experimental configurations without extensive modification of the pumps or pump calibrations. Their accuracy is comparable to commercial and DIY alternatives[11,50] and our GUI precisely controls pump speeds when handling flow sensitive samples. For additional flexibility, PHIL can utilize any stepper motor-driven pump by using the pump calibration tool implemented in our GUI. Improved pumping resolution while pipetting small volumes can be achieved through a variety of methods e.g. by utilizing small diameter hydrophobic tubing as pipets or open-source syringe pumps which include geared stepper motors. While these small diameter tubes can improve pipetting accuracy, they reduce the maximum achievable flow rate and increase the net cost of PHIL production significantly. Additionally, the miniaturized nature of the robot also allows to chain many of them together for more complex workflows. PHIL will also be useful for many other applications not tested here. For example, the modular design of the pipets and robot arm also allows the inclusion of novel actuators including LEDs for optogenetic experiments, electromagnets for magnetic particle separation experiments, anode/cathode pairs for electroporation, or direct mechanical stimulation of artificial tissues. The incubation chamber can be rapidly modified to include other devices such as tube holders for tip rinsing or automated sample aliquoting with minimal design changes.

PHILs potential applications are not limited to the examples explored here. Within a temperature-controlled enclosure and connected to external gas and humidity sources, PHIL can independently culture sensitive cell types. These include simulated co-cultures by transferring fluids between wells, dividing single cultures into multiple wells for separate treatments, real-time automated cell culture interventions based on observed cell culture phenomena, and the sequential staining of fixed samples over many cycles without sample loss or displacement. PHILs versatility and open-source design make it ideal for exploring small-scale experimental protocols without large cost, space, and time investment in a wide range of experimental laboratories.

## Methods

**Ethical statement**. The research presented here complies with all relevant ethical regulations. Animal experiments were approved according to Institutional guidelines of ETH Zurich and Swiss Federal Law by the veterinary office of Canton Basel-Stadt, Switzerland (approval no. 2655). Anonymized human UCB samples were collected from healthy newborns of both sexes at the University Hospitals Basel or Zurich (Department of Obstetrics, University Hospital Zurich and Triemli Hospital, Zurich, Switzerland) with parental informed consent for published research, and without participant compensation. Relevant ethical regulations were followed, according to the guidelines of the local Basel ethics committees (vote 13/2007V, S-112/2010, EKNZ2015/335) or the ethics boards of the canton Zurich (KEK-StV-Nr. 40/14).

**PHIL design and fabrication**. Using AutoCAD 2018 (Autodesk Inc.) we designed a 3D printable 5-bar planar robot and printed all components on an Ultimaker 2 extended+ 3D printer (Part number: UM2EXTPLUS, Ultimaker B.V. Netherlands) with 3Djake niceABS black (niceshops GmbH). Slicing and gcode generation was done in Cura (Ultimaker B.V. Netherlands) using generic ABS settings. Models were printed with a 0.4 mm nozzle, 0.15 mm layer height and 18% infill except the arm mounting brackets which were printed with 100% infill. Briefly, all printed parts were assembled into the final product using standard 8, 10, and 16 mm M3 flathead screws to attach motors and printed parts. Stepper motors on the main body were 23 mm NEMA 17 stepper motors (part number: 17HS0423). Peristaltic pumps were driven by 60 mm NEMA 17 stepper motors (part number: 17HS6002). Each motor is driven by a dedicated stepstick DRV8825 stepper driver and powered by a 24 V 15 A power supply connected to the Arduino Mega Shield. Limit switches were installed on the main body and robot arms using 10 mm M1.5 screws. The 24 V 15 A power supply is mounted in a 3D printed enclosure along with an Arduino Mega 2560 Rev3 (Arduino, A000067) and custom shield with 10 mm M4 flathead screws. Custom Arduino shields utilized in all experiments were designed in Autodesk EAGLE 2018 (Autodesk Inc.) and produced in house, while 3 of the PHIL robots constructed by Ph.D. students were assembled by an external manufacturing firm (https://www.pcbway.com/project/shareproject/ArduinoMega_14Stepper_V3.html, ~20 USD each). Each pipet was connected to its corresponding pump using 0.56 × 1.07 mm PTFE-tubing (Thermo Fisher, W39241). Pipets were produced buy bending 0.6 mm × 0.3 mm × 25 mm (OD × ID × L) stainless steel tubing (New England Small Tube Corp. NE-1310-03) in a custom 3D printed bending apparatus. A complete set of production and assembly instructions are available online (https://bsse.ethz.ch/csd/Hardware/PHILrobot.html, https://github.com/CSDGroup/PHIL).

**PHIL control**. Our graphic user interface was implemented in MatLAB (The MathWorks Inc.) as a single script. Briefly, we programmed a simple model of our device using the Robotics System Toolbox (The MathWorks Inc.) which, when given a single XY position within the working area of our device, generates two angular rotations for the primary arms to position the pipets in the desired location. These rotations are then compared to the current configuration of the arms and the desired step counts and directions are sent to the Arduino via serial communication. The Arduino then implements those steps via a custom script written using the Accelstepper Arduino library (https://www.airspayce.com/mikem/arduino/AccelStepper/index.html). Our GUI and Arduino script as well as instructions for their implementation are available online (https://bsse.ethz.ch/csd/Hardware/PHILrobot.html, https://github.com/CSDGroup/PHIL).

**PHIL motion calibration**. Pipet positioning accuracy was conducted by filling a single pipet with 100 µM FITC (F6377-100G, Sigma-Aldrich) in PBS and positioning it in the center of well F12 in a 96-well plate (Greiner-CELLSTAR®-96-Well) approximately 0.1 mm above the bottom of the well. FITC fluorescence within the pipet as it moved away from this location and returned was imaged every 1 s on a Nikon Ti-Eclipse inverted microscope that was equipped with an Orca Flash 4.0 camera (Hamamatsu Photonics K.K.), using a ×10 objective (NA 0.45) (Nikon Instruments Europe B.V.) and pipet location was recorded using custom scripts written for Youscope (www.youscope.org).

**Pipet surface treatment**. Pipets were treated with oxygen-plasma at 15 W for 2 min and wetted with 1% TMCS. Pipets were then placed on a hot plate held at 90 °C until dry.

**Cleaning PHIL**. PHIL was cleaned between experiments using a preprogrammed script. Briefly, a six-well plate containing 6 mL FACS Clean (Thermo Fisher, BD 340345), 6 mL FACS Rinse (Thermo Fisher, BD 340346), 6 mL 80% ethanol, and 6 mL diH₂O in separate wells was loaded into the working area. PHIL then pipetted 2 mL per pipet from each well by running the corresponding pump in reverse. After cleaning each pump was run in reverse until all Teflon tubing was dry.

**Pump calibration**. Pipet tips were primed by pumping liquids into them until a small volume was ejected. Pumped volume accuracies were determined by pumping 10, 25, 100, and 1000 µL of diH₂O onto the walls of empty 1.5 mL MaxyClear snaplock microcentrifuge tubes (MCT-150-A, Corning Inc.) which were weighed before and after to determine the true pumped volume ($n = 10$ repetitions per condition). Maximum flow rate calculations were determined by pumping 100 µL diH₂O onto the walls of empty PCR tubes at 10, 25, 50, and 100 µL/s which were weighed before and after to determine true volume pumped. Automated FITC dilution curves were generated by loading pipets with 100 µM FITC in PBS and PBS before pipetting 0–100 µM FITC solutions in 96-well plates (Greiner-CELLSTAR®-96-Well) in 10 µM increments. FITC fluorescence was imaged on a Nikon Ti-Eclipse inverted microscope equipped with an Orca Flash 4.0 camera (Hamamatsu Photonics K.K.), using a ×10 objective (NA 0.45) (Nikon Instruments Europe B.V.).

**Dynamic media exchange**. PHIL pumps connected to PBS and 100 µM FITC (Sigma-Aldrich, F6377-100G) in PBS filled 96-well plate (Greiner-CELLSTAR®-96-Well) wells with 100 µL PBS. PHIL then aspirated the well contents, added 100 µL 100 µM FITC, aspirated, and added 100 µL PBS over various periods and intervals. All volumes were pumped at 50 µL/s. FITC fluorescence was imaged on a Nikon Ti-Eclipse inverted microscope equipped with an Orca Flash 4.0 camera (Hamamatsu Photonics K.K.), using a ×10 objective (NA 0.45) (Nikon Instruments Europe B.V.).

**Human hematopoietic stem and progenitor cell isolation**. Human cord blood cells were processed by density gradient centrifugation, and CD34⁺ cells were isolated using EasySep™ CD34 positive selection kit II (samples from Basel, Cat# 17896, Stemcell Technologies, Vancouver, BC, Canada).

**Human hematopoietic stem and progenitor cell cultures**. HSPCs were cultured as previously described[21]. Cells were cultured in phenol red free Iscove's modified Dulbecco's medium (IMDM, Gibco #21056-023) supplemented with 20% BIT (Stemcell Technologies), 100 ng/mL human recombinant Stem Cell Factor (SCF, #255-SC), 50 ng/mL human recombinant Thrombopoietin (TPO, #288-TP), 100 ng/mL human Fms-related tyrosine kinase 3 ligand (Flt3L, #308-FK, all R&D Systems), 2 mM L-GlutaMAX (Gibco #35050-038), 100 U/mL penicillin and 100 µg/mL Streptomycin (Gibco #15140-122), and 4 µg/mL low density lipoproteins (Stem Cell Technologies #02698), 55 µM 2-mercaptoethanol (Gibco). Cells were fixed on µ-Slide VI⁰·⁴ (#80606, Ibidi GmbH) slides coated for 1 h with 10 µg/mL anti-CD43-biotin (MEM-59, ExBio) in PBS. Washing steps were conducted with PBS.

**ESC culture**. R1 NanogVenus cells were routinely cultured in SerumLIF media on 0.1% gelatin (Sigma-Aldrich, G1890-100G) coated plates[51,52]. SerumLIF media consists of 10% FCS (PAN, P30-2602), 10 ng/mL LIF (Cell guidance Systems, GFM200-1000), 2 mM GlutaMAX (Thermo Fisher, 35050-038), 1% non-essential amino acids (Thermo Fisher, 11140-035), 1 mM sodium pyruvate (Sigma-Aldrich, S8636) and 50 µM β-mercaptoethanol (Sigma-Aldrich, M6250) in DMEM (Thermo Fisher, 11960-085) basal media. E cadherin (Primorigen Biosciences, S2112-500UG) was added at 0.4 µg/100 µl PBS++ per well for coating and incubated for 1 h at 37 °C. Cells were seeded in SerumLIF in 96 well plates (Ibidi, 89626) coated with E-cadherin prior to stimulation experiments. The media during imaging is the same as routine culture media except basal media is phenol red free DMEM (D1145). ESC culture experiments were conducted on environmentally controlled microscopes maintained at 37 °C using a feedback-controlled heating unit (Life Imaging Services) installed in aluminium-slot frame boxes lined with black, foamed insulation-boards. Humidity (>98%) and gas composition (5% CO₂,

5% O₂, 90% N₂, PanGas) were maintained by constantly flowing 1 mL/h mixed gas into PHIL through a previously reported 3D printed temperature-controlled air humidifier[23] containing roughly 100 mL of diH₂O at 37 °C (https://bsse.ethz.ch/csd/Hardware/3DPHumidifier.html). Relative humidity was recorded using a data logger (Sensirion, SHT31-DIS-B).

**Mice**. Experiments were conducted with 12–16-weeks old male GATA2VENUS and GFP-p65/H2B-mCHERRY mice. Trangenic mice were previously published[31]. Mice were acclimatized for at least 1 week before the start of an experiment. Mice were supplied with environmental enrichment and housed with an inverse 12 h day–night cycle in a temperature (21 ± 2 °C) and humidity (55 ± 10%) controlled room with ad libitum access to standard diet and drinking water at all times in individually ventilated hygienic cages with 2–5 mice per cage. The general well-being of the mice was routinely monitored by animal facility caretakers by daily visual inspections. Mice were euthanized if symptoms of pain and/or distress were observed. Mice were randomly assigned to experimental groups and in some cases pooled for experiments to reduce biological variability. Animal experiments were approved according to Institutional guidelines of ETH Zurich and Swiss Federal Law by veterinary office of Canton Basel-Stadt, Switzerland (approval no. 2655).

**Murine myeloid progenitor and GMP isolation**. Primary cells were isolated and sorted as previously described[31,53]. In brief, femurs, tibiae, coxae, and vertebrae were isolated and crushed in PBS (2% FCS, 2 mM EDTA) and filtered through a 40-µm nylon mesh. Erythrocytes were lysed for 3 min on ice in ACK lysing buffer (Lonza), stained with biotinylated lineage antibodies against CD3ε (145-2C11), CD19 (eBio1D3), TER-119 (TER-119), B220 (RA3-6B2), Ly-6G (RB6-8C5) and CD11b (M1/70), and labelled with streptavidin-conjugated magnetic beads (Roti-MagBeads, Roche) before immuno-magnetic depletion. For isolation of GFP-p65 GMPs, cells were stained with CD16/32-PerCp-Cy5.5 (Biolegend, Clone 93), Sca1-Pacific Blue (Biolegend, Clone D7), cKit-BV510 (Biolegend, Clone 2B8), CD150-BV650 (Biolegend, Clone TC15-12F12.2), streptavidin-BV711 (BD Biosciences), CD34-eFluor660 (Invitrogen, Clone RAM34) and CD48-APCeFl780 (Invitrogen, Clone HM48-1). For isolation of GATA2VENUS myeloid progenitors, cells were stained with CD16/32-APCCY7 (Biolegend, Clone 93), Sca1-PacBlue (Biolegend, Clone D7), cKit-BV711 (BD Biosciences, Clone 2B8), CD150-BV650 (Biolegend, Clone TC15-12F12.2), streptavidin-BV570 (Biolegend), CD41-PerCPeFL710 (eBioscience, Clone MWReg30) and CD105-APC (Biolegend, Clone MJ7/18). Cells were stained for 90 min on ice and sorted using a BD FACS Aria I or III with 70-µm nozzle, single-cell purity mode and sorting purities ≥98%.

**GMP culture**. Primary murine hematopoietic cells were isolated from GFP-p65/H2B-mCHERRY transgenic mice described above[43,44]. Cells were cultured before and during time-lapse movies in 4-well micro-inserts (Ibidi) within a 24 well plate with glass bottom (Greiner) for 12 h. Before the start of cultures, the plate was coated with 10 µg/mL anti-CD43-biotin antibody for 2 h at room temperature in order to reduce cell movement[32,40]. After washing with PBS, 1 mL of IB20/SI media (IB20 = custom IMDM without riboflavin (Thermo Fisher) supplemented with 20% BIT (Stem Cell Technologies), 50 U/mL Penicillin, 50 µg/mL Streptomycin (Gibco), GlutaMAX (Gibco), 2-Mercaptoethanol (50uM, Gibco); SI = 100 ng/mL murine SCF + 10 ng/mL murine IL-3 (both Peprotech)) was added per well and cells were cultured at 37 °C and 5% CO₂. For GMP displacement experiments all GMPs were imaged for 1 h before half the wells were washed with 1 mL base media at 50 µL/s. Cells were manually tracked and changes in motility in washed and unwashed wells determined using R.

**Confocal time-lapse imaging**. Time-lapse experiments were conducted at 37 °C, 5% O₂ and 5% CO₂ with detailed media conditions described above using a Nikon A1 confocal microscope. Images were acquired using the blue (GFP, 3.5% intensity) and green (mCherry, 1% intensity) laser, custom GFP and mCherry emission filter settings and a 20× CFI Plan Apochromat λ objective. During time-lapse movies, images were acquired every 9 min for 12 h.

**Image quantification and analyses**. 12-bit images with 2048 × 2048-pixel resolution were saved in.tiff format and linearly transformed to 8-bit using channel-optimized white points. Nuclear segmentation was accomplished using images of the mCherry channel and fastER[33]. Trained labelling masks were eroded (settings: dilation −2) to ensure that the segmented nuclei contained no cytoplasm. Tracking and quantification of fluorescence channels was accomplished as previously described[54], and analysed using R 3.4.2 (R-Project). Time series of each channel were generated using mean intensities of the segmented areas. Data from all cells were organized in a single data frame. In order to account for morphological artifacts across time, GFP (p65) time series were normalized by mCherry (H2B) time series and the resulting time series were scaled such that the mean of the baseline (first 6 timepoints before stimulation) was equal to 1 for all cells.

**Generation and analysis of single-cell trajectories**. The sum of nuclear fluorescence intensity of GFP-p65 was divided by the sum of fluorescence intensity of H2B-mCHERRY for each cell at each timepoint, with the assumption that the sum

of H2B fluorescence intensity stays constant for the imaging period (720 min). Fold changes were then calculated by normalizing to the mean of baseline (0–60 min) values before growth factor stimulation. The resulting trajectories were mirrored along the horizontal axis to intuitively reflect p65 activity and a value of 2 was added to all timepoints in order to re-set the mean of baseline to 1 for each trajectory. Log2 transformed values were plotted.

**Blind manual time series classification**. For "manual" classification, time series were assigned to the predefined categories non-responsive, sustained, transient, oscillatory and unclear/outlier by the experimenter. Time series were chosen randomly and displayed without any meta data (e.g. stimulation information) to the experimenter by an algorithm implemented in R to assure that there was no classification bias. All series of all experiments of the GMP data set were classified in a single session.

**Automated immunostaining of GATA2VENUS in myeloid progenitors**. Immunostaining of VENUS fluorescent protein in myeloid progenitors was performed according to protocols as described[31,34,39,55]. Briefly, different subsets of myeloid progenitors based on VENUS expression were sorted by FACS (BD ARIA III), directly seeded on (10 µg/mL) anti CD43-biotin antibody-coated plastic-bottom 96-well plates (Greiner Bio-one), stored for 30 min at 4 °C and fixed with 4% paraformaldehyde (Sigma Aldrich) for 10 min at RT. The automated washing procedure conducted by PHIL consisted of 3× washing step with 100 µL PBS, 2× permeabilization steps with 100 µL TBS-T (0.1% Tween (Sigma Aldrich) + 0.2% triton-X (Applichem) in TBS buffer), 1 h incubation at RT with 100 µL blocking buffer (10% donkey serum (Jackson Immuno research) in TBS-T), incubated with 100 µl of 4 µg/mL chicken-anti GFP primary antibody (Aves) in blocking buffer overnight at RT, washed 3× with 100 µL TBS-T, stained with 100 µL donkey-anti chicken Alexa 488 secondary antibody (10 µg/mL dilution in TBS-T) for 1 h, incubated with 100 µL DAPI (1:1000 in blocking buffer), washed 3× with 100 µL TBS-T, followed by final incubation in TBS before imaging. Images were acquired on a Nikon Eclipse Ti-E microscope using Lumencore light source with 0.7× camera adapter and ×10 objective with 0.45 NA (Plan Apo) and analysed using bioimaging pipeline described below.

**Bioimaging pipeline for image acquisition, detection and quantification**. Fluorescence images of immunostained myeloid progenitors were acquired on Nikon Eclipse Ti-E microscope in an automated manner using custom written software. To quantify the signal of anti-VENUS immunostaining in single cells, background signal was normalized by using BaSiC tool as previously described[56] and segmentation of cells was performed on DAPI signal. Finally, quantification of anti-VENUS immunostaining signal was performed using fastER segmentation tool as described[33]. Data was analysed using custom written R scripts.

**Tissue preparation for automated immunostaining**. Murine bone marrow were sectioned as described before[57]. Brain was prepared in the same way, only without decalcification. Briefly, tissues were fixed in freshly prepared 4% methanol-free formaldehyde solution (Thermo Scientific, Cat# 28906) for 24 h directly after dissection. Bones were submerged in 10% EDTA, pH 8 (BioSolve, Cat# 0005142391BS) for 14 days to decalcify. Tissues were embedded in 4% low gelling temperature agarose (Merck, Cat#A0169) and sectioned by vibratome (Leica VT1200 S) into 100–150 µm-thick sections.

**Automated immunostaining tissues with PHIL**. Automated immunostainings were performed at room temperature in double side adhesive silicon chambers (Merck, Cat# GBL620003) attached on superfrost glass slides (VWR). Samples were fixed, bones decalcified, sectioned, and placed on glass slides as previously described[57]. Tissue sections were blocked and permeabilized with TBS (0.1 M Tris, 0.15 M Nacl, pH 7.5) containing 10% donkey serum (Jackson ImmunoResearch, 017-000-121), 0.05% Tween-20 (Merck, P9416) and 0.1% Triton X-100 (PanReac AppliChem, A1388) for 2 h before addition of 200 µL primary antibody solutions. Bone primary antibody solution contained, goat anti-Collagen 1 IgG (Southern Biotech, 1310-01), rat anti-CD41 IgG (eBioscience, 16-0411-85), and DAPI (Merck, D9542) to visualize bone surface, megakaryocytes and cell nucleus respectively. Cerebrum primary antibody solution contained mouse anti-GFAP IgG (R&D Systems, MAB2594) and DAPI to visualize astrocytes and cell nucleus respectively. Sections were incubated in primary antibodies for 15 h before 4 wash cycles with 200 µL washing solution composed of TBS containing 0.05% Tween-20. Sections were then incubated in 200 µL secondary antibody solution for 2 h. Bone secondary antibody solution contained Alexa Fluor 647 donkey anti-goat IgG (Invitrogen, A-21447) and Alexa Fluor 555 donkey anti-rat IgG (Invitrogen, A-48270). Cerebrum secondary antibody solution contained Alexa Fluor 546 donkey anti-mouse IgG (Invitrogen, A-10036). Secondary antibody solutions were washed away using 200 µL washing solution as above. Primary and secondary antibodies were diluted in the same blocking buffer described above. Sections were mounted in homemade mounting medium (80% Glycerol (PanReac AppliChem, A1123) in TBS containing 0.2 M N-propyl gallate (Merck, 02370), pH 8.5) with a size 1.5 coverslips (TedPella).

**Confocal microscopy for immunostained tissues**. Images were acquired on a Leica TCS SP8 confocal microscope using Leica type G immersion liquid with a ×63 glycerol immersion lens (NA 1.3, FWD 0.28 mm). The scans were performed at 400 Hz, room temperature, in bidirectional mode at 1024 × 1024 pixel resolution. Only HyD detectors were used for signal acquisition.

**Statistics and reproducibility**. All analyses were performed in RStudio (version 4.1.2). No statistical methods were used to predetermine sample size. Individual statistical tests performed are mentioned in the figure legends; in general, non-parametric tests were performed. No data were excluded from the analyses. The experiments were not randomized. The Investigators were blinded to allocation during live cell stimulation outcome assessment.

**Reporting summary**. Further information on research design is available in the Nature Research Reporting Summary linked to this article.

## Data availability
All data are provided in the included Article, Supplementary Information, and Source Data files. 3D printable files, assembly instructions, and PHIL control software are provided in the included Supplementary Software file.

## Code availability
PHIL follows open-source software and hardware standards. The steps to assemble a complete functioning system and corresponding software are described online (https://bsse.ethz.ch/csd/Hardware/PHILrobot.html, https://github.com/CSDGroup/PHIL, https://doi.org/10.5281/zenodo.6402030).

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

## Acknowledgements

We thank the D-BSSE single cell facilities for technical support. This work was supported by Swiss National Science Foundation grants 179490 to T.S.

## Author contributions

P.D. designed and implemented the robot hardware, control software, and environmental control apparatus. T.S., N.A., F.S., T.K., and Y.Z. provided feedback improving the functionality and user experience of the system. Y.Z. produced murine tissue staining data, N.A. myeloid progenitor immunostaining data, F.S. cord blood washing data, and G.A. embryonic stem cell culture data. A.W., D.S., Y.Z., and F.S. provided feedback on the robot assembly process. S.K. provided biological samples. T.S. and D.L. developed continuous quantitative single-cell imaging and maintained it with T.K.. T.S. designed and supervised the project.

## Competing interests

The authors declare no competing interests.
