## [Peer Review File · Nature Communications]

Reviewers' Comments:

Reviewer #1:

Remarks to the Author:

The work presented by Timm Schroeder and coworkers shows the design, construction, and potential utilization of the Pipetting Helper Imaging Lid (PHIL), a peristaltic pump-based liquid handling system for microscopy applications. The authors claim to have created the first design of a pipetting robot based on a "5-bar planar arm" traverse, and in fact, this design was unknown to me from past open-source and commercial systems. From an open laboratory engineer's point of view, there is nothing to complain about this design, but there is some room for improvement in the presentation of the open-source code and some aspects of the pipetting performance.

In general, the article is clearly structured, uses appropriate language, and, most importantly, provides easy-to-understand instructions for rebuilding the design, so after successful revision, I would like to make a recommendation in favor of publication.

The mere description of this new design would seem to be of rather little relevance to the research field (microscopy automation), but the detailed construction instructions (incl. videos) make this article quite interesting for the readership of Nature communications. The described applications to microscopy seem to me convincing that PHIL is suitable to promote this field of research. By adapting some parts of the source code (see comments below), it might be possible to attract even more readers outside this target group.

Comments and suggested improvements:

-The introduction to this manuscript gives a brief but not too sparse overview of the publication landscape on open-source liquid handling systems; however, the classification of the current design does not really succeed in the course of this introduction. The scope of the described system (microscopy and cell culture) is mentioned, but it is not classified which type of design is presented. A few key points in the last section of the introduction, such as peristaltic pump, focus of the design on the loading of well plates, etc., would help to categorize the design in the literary landscape without anticipating the results. Likewise, the keyword "peristaltic pump" should already be mentioned in the abstract.

-The design is claimed to be open-source which is supported by the source code being made freely available for reproduction. However, I was wondering about the licensing of this design. To my knowledge, the publication at Nature Communications is released under a CC-BY license. However, much of the source code for reproducing the design (Stl-files) is hosted on a server owned by the authors and thus can be modified even after publication. So, in my legal opinion, this part of the source code might not be part of this publication. On the authors' website (<https://bsse.ethz.ch/csd/Hardware/PHILrobot.html>) I could not find any hint about licensing of the design. If you do not use an open-source license, anyone who contributes to your project also gets exclusive copyright. This would mean that really nobody would be allowed to use, copy, distribute, or modify the contributions. Not even you. I recommend the Open-source Hardware Association (<https://www.oshwa.org/>) as a place to visit. There you can even get your design certified as "open-source" for free (<https://certification.oshwa.org/>).

-To ensure source code longevity, I would also like to see authors make their design available on a repository, which 1.) guarantees indefinite availability and 2.) does not allow subsequent changes to the source code state after the publication date (e.g., Mendeley Data, the Open Science Framework, and Zenodo).

-Furthermore, in the introduction, existing open-source liquid handling systems are criticized for being suitable only for specialized applications. However, in my opinion, the design presented here is also strongly specialized for specific microscopic applications. This is not a problem so far, because according to the common definition of open-source hardware, this type of hardware should have the feature of modifiability and users should be able to adapt the design according to their needs. Here I see potential for optimization in the presented study: the authors only provide Stl-files for 3D printing, so that customization of the design will be exhausting. I find the design very interesting and have ideas about how I could use parts of the design for applications that suit my research (e.g., modifying the workspace beyond well-plates). For this, the community would need the native CAD-files, which is why I ask to make them part of the available source code. I am sure the authors are aware of this as they have also described possible modifications "LEDs for optogenetic experiments, electromagnets for magnetic cell separation experiments, or anode/cathode pairs for electroporation".

-The authors report predominantly good benchmarks for their system: No cross-contamination and

good pipetting performance for high volumes (>25 µL), low cell loss when washing flow-sensitive cells, no differences to cultivation in an ordinary incubator. However, 20% gravimetric error at 10 µL seems to be too much for practical utilization. 10 µL is still a common volume for working in 96-well plates. I am sure that the pipetting performance can be improved here. As I understand it, this error was determined as part of the calibration of the pipette. What is not quite clear to me is how the liquid delivery takes place during calibration (and also during filling of the plate): Was the liquid pipetted into the air, onto the wall, or into a pre-loaded liquid? Was calibration performed in the same way as pipetting into the plates later? Possibly the authors should use a finer pipette tip for this and pipette below the liquid surface. The 23-gauge needle used as pipette is not really the smallest option possible. With direct displacement liquid transfer, there must not be any air cushion in front of or behind the dispensed liquid. How was the pipette tip primed before performing a liquid transfer?

-In general: How deep does the pipette tip dip into the reservoir during aspiration and liquid delivery? Is this variable for different applications? Such information should best be explained at the beginning of the result section as part of the system design. Further, how can this be influenced by the user? Is the pipette tip lowered simultaneously during aspiration so that it always aspirates just below the liquid surface to avoid turbulence? Some commercial devices ensure this by using conductive tips. In my experience, this can be solved by the calculated volume + a buffer height (Z-position defined by the filling height of the plate).

-I am not aware of any ISO guideline for the calibration of peristaltic pumps in the specific scope presented here. However, for the applications described here, usually, air displacement and direct displacement pipettes are used, which are regulated according to ISO8655. Authors are encouraged to consult this or another appropriate guideline and clarify the description of the actual pipetting procedure/calibration.

-At the beginning of the result section (design chapter), some other important parameters of the system seem to be missing, which would be important for a technically interested reader: What is the maximum z-stroke of the pipetting arm? The dimensions of the pipette and the PTFE tubing might also be mentioned here. How much µL volume one micro step of the motors corresponds to would be useful to know. In fact, we were able to achieve excellent pipetting accuracy on a syringe pump in the sub-microliter range by using a NEMA motor equipped with an additional gearbox.

-A maximum height of 45 mm for target vessels limits the applications of the system but seems ok as a compromise for a short optical path. For flat vessels other than well-plates, however, 45 mm seem rather small. Other users may want to rinse the pipette in external rinsing vials, then you do not have to rinse into unused wells of the well-plate.

-The cost mentioned in the manuscript (550€) differs from the order list (719€). Where does this difference come from? The order list seems to be incomplete as well, I could not find items like the microcontroller, PCB shield, tubings, connectors, etc. I also could not find the digital PCB layouts online as proclaimed in the article. Additionally, a technical circuit drawing would be useful (preferably created with open-source software).

Reviewer #2:

Remarks to the Author:

The authors present exciting technology in the form of a DIY liquid handling robot that can perform precise and sensitive liquid manipulations on a microscope stage. This development will likely be of interest to many in the biosciences. The paper is well-written and the characterizations seem thorough. I recommend publication after clarification of a few minor items, below. Congratulations to the authors – I hope my students build and use one for themselves.

-The authors mention that PHIL can be used as an environmental chamber on its own with simple connections to preconditioned air, but there is no data to support this. This is not a trivial claim and requires good temperature monitoring possibly with feedback, airtight material and connections, etc. In our experience even expensive stage-top incubators can regularly fail to support cell viability beyond ~24 hrs for one of number of reasons. Can the authors provide data that PHIL can adequately serve as an incubation chamber? If not, then perhaps its use as such should be left as speculation and the text should focus more on its use in controlled environments (eg a large environmental chamber that surrounds the scope and PHIL).

-on first read, I thought the 3-10 reusable pipettes were in a multichannel setup that allowed parallel manipulations in separate wells, rather than a bundle that all went into one well. Perhaps the fact that you have up to 10 channels of control in each well could be better emphasized to make this point more clear for the reader. This was also made confusing by varying descriptions (descriptions include: 1-10 reusable pipettes; can dispense through 1 to 9 separate tube channels; pumps can drive flow through 3-10 pipettes). Why the variable descriptions? This could be clarified

-The expt in 3b could be better explained. Is the Venus FACS panel data from a flow cytometer, or from a microscope after sorting? The text states that the distributions between the left and right plots are the same, but they are not (eg "high" overlaps with "mid" on right plot, but not on left).

-The cell loss results are very impressive! It would be good to have more details of how the washes are done. For example, are these full washes where all of the liquid is removed? Was there just one wash? Does aspiration speed matter? Does positioning of the aspirator matter (eg in the middle of the well vs the wall). Does positioning of dispensing matter? If these parameters haven't been exhaustively tested, at least the authors could provide these details for the data presented.

-In fig 5D, it would be good to know how many cells were counted for each condition.

Reviewer #3:

Remarks to the Author:

The paper presents an open-source liquid handling robot for cell culture. Its main features are low cost, easy to assemble by non-trained persons and a small footprint that allows the robot to be placed in microscopes and other hardware. The robot has been carefully described and characterized and supplementary materials provide detailed assembly instructions and technical details. The robot has also been tested in simple experiments to characterize cross contamination and reliability. Finally, two real experiments have been carried out (automated immunostaining and automated pipetting of long-term live cell cultures). Both experiments have been compared with experiments carried out by humans, achieving similar performance. The paper is clear and well written.

Open-source laboratory equipment have been published several times before, specially using 3D printing techniques. The main motivations are reducing costs, making lab equipment affordable and satisfying special requirements. However, most of the papers lack of real validation tests or the manufacturing requires advance skills. In this regard, this paper provides an outstanding contribution as provides an easy to build liquid handling robot which has been validated thoroughly in complex experiments. Moreover, the versatility of the liquid-handling robot and its low footprint make it suitable to produce new kinds of experiments that could have not been possible before. In my experience, persons without a technical background have issues assembling these types of systems. Overtightening screws, bad soldering or wiring issues can be difficult to debug. In this paper, they have avoided crimping operations (for connectors) by buying motors with pre-crimped connectors and soldering male headers directly to cables. This causes that sourcing the stepper motors becomes more difficult (most manufactures do not provide connectors in the motors) and requires soldering skills. Soldering can be a difficult task without help or training and a bad soldering technique can produce bad contacts which are difficult to debug as they can produce intermittent behaviors. Thus, I am a bit concern about the quality of robots built by persons without previous training in mechatronics. While the paper states that 5 robots were built by biology PhD students, I would appreciate if the discussion was expanded with the challenges that they found and if they needed help. Additionally, it seems that the PCB was already soldered the video that shows the assembly of the robot. Please, specify if this was taken into account in the time required for the assembly and if the biologists also soldered the PCBs. Finally, the stepper drivers have a potentiometer, which controls the amount of current that flows through the motor, that needs to be adjusted. I could not find any instruction about how to adjust it in the assembly instructions.

In order to increase the use of the robot, it would be nice if the authors upload the STL files to some online 3D printing services. Thus, persons who want to build the robot would not require access to a 3D printer. The same could apply to the PCB board, which seems the most challenging step for me (e.g. Macrofab, Aisler or Seeed Studio offer manufacturing and assembling of PCBs).

These actions could decrease the difficulty to build the robot, increasing the impact of the paper. I found strange that the control software of the open-source robot was designed in Matlab, which is a commercial product. This limits the use of the robot to laboratories with a license or they would need to create their own high-level control, which requires specialized knowledge. Have you considered using GNU Octave instead?

While the paper focuses on cell cultivation, I wonder if it could be applied to other kinds of experiments. I guess that DNA or bacteria experiments are probably out of scope of the machine as avoiding cross contamination is critical in these kinds of experiments.

Reviewer #1 (Remarks to the Author):

Thank you for your time and your detailed and helpful suggestions!

General information:

In response to our reviewers and to the feedback from users who have already started producing PHILs in other laboratories following our BioRxiv publication, we have made several modifications to the provided online resources. These include:

- Creating a github.com repository containing:
 - Licensing information.
 - All raw files (.dwg) for PHILs design.
 - 3D printable files (.stl) for PHILs production.
 - User interface files (.m) for PHILs user interface.
 - Printed circuit board files for producing PHILs Arduino shield.
- Providing increased detail regarding purchasing PHIL components including a list for purchasing a basic version of PHIL and a list for an advanced version of PHIL.
- Improving available PHIL construction instructions.

We have also updated our website and manuscript to include these improved resources. We will continue to update our website and github repository in response to relevant user feedback (while always also keeping the previous versions available).

The work presented by Timm Schroeder and coworkers shows the design, construction, and potential utilization of the Pipetting Helper Imaging Lid (PHIL), a peristaltic pump-based liquid handling system for microscopy applications. The authors claim to have created the first design of a pipetting robot based on a "5-bar planar arm" traverse, and in fact, this design was unknown to me from past open-source and commercial systems. From an open laboratory engineer's point of view, there is nothing to complain about this design, but there is some room for improvement in the presentation of the open-source code and some aspects of the pipetting performance.

In general, the article is clearly structured, uses appropriate language, and, most importantly, provides easy-to-understand instructions for rebuilding the design, so after successful revision, I would like to make a recommendation in favor of publication.

The mere description of this new design would seem to be of rather little relevance to the research field (microscopy automation), but the detailed construction instructions (incl. videos) make this article quite interesting for the readership of Nature communications. The described applications to microscopy seem to me convincing that PHIL is suitable to promote this field of research. By adapting some parts of the source code (see comments below), it might be possible to attract even more readers outside this target group.

Comments and suggested improvements:

The introduction to this manuscript gives a brief but not too sparse overview of the publication landscape on open-source liquid handling systems; however, the classification of the current design does not really succeed in the course of this introduction. The scope of the described system (microscopy and cell culture) is mentioned, but it is not classified which type of design is presented. A few key points in the last section of the introduction, such as peristaltic pump, focus of the design on the loading of well plates, etc., would help to categorize the design in the literary landscape without anticipating the results. Likewise, the keyword "peristaltic pump" should already be mentioned in the abstract.

Thank you. We have updated the abstract and introduction.

Abstract:

“PHIL successfully automated pipetting (incl. aspiration) for e.g. tissue immunostainings and stimulations of live stem and progenitor cells during time-lapse microscopy using 3D printed peristaltic pumps.”

Introduction:

“The Pipetting Helper Imaging Lid (PHIL) is small enough to be installed almost anywhere, including on most microscope stages, and automates complex liquid handling sequences in standard well plates while also allowing conditioning of the incubation gas atmosphere.”

The design is claimed to be open-source which is supported by the source code being made freely available for reproduction. However, I was wondering about the licensing of this design. To my knowledge, the publication at Nature Communications is released under a CC-BY license. However, much of the source code for reproducing the design (Stl-files) is hosted on a server owned by the authors and thus can be modified even after publication. So, in my legal opinion, this part of the source code might not be part of this publication. On the authors' website (<https://bsse.ethz.ch/csd/Hardware/PHILrobot.html>) I could not find any hint about licensing of the design. If you do not use an open-source license, anyone who contributes to your project also gets exclusive copyright. This would mean that really nobody would be allowed to use, copy, distribute, or modify the contributions. Not even you. I recommend the Open-source Hardware Association (<https://www.oshwa.org/>) as a place to visit. There you can even get your design certified as “open-source” for free (<https://certification.oshwa.org/>).

To ensure source code longevity, I would also like to see authors make their design available on a repository, which 1.) guarantees indefinite availability and 2.) does not allow subsequent changes to the source code state after the publication date (e.g., Mendeley Data, the Open Science Framework, and Zenodo).

Thank you, these comments are highly appreciated. We have now also created a public repository on github.com that provides licensing information under a standard MIT license, a permanent history of modifications to the code, and guarantees independent availability. OSHWA.org was indeed a valuable resource for us when selecting the appropriate license, but we did not see the benefit of undergoing their certification process. The link to this public github repository can now be found in section “Production and assembly of PHIL robots,” with the following statement:

“Detailed part lists and instructions for non-experienced users are available online (<https://bsse.ethz.ch/csd/Hardware/PHILrobot.html>, <https://github.com/CSDGroup/PHIL>).”

Furthermore, in the introduction, existing open-source liquid handling systems are criticized for being suitable only for specialized applications. However, in my opinion, the design presented here is also strongly specialized for specific microscopic applications. This is not a problem so far, because according to the common definition of open-source hardware, this type of hardware should have the feature of modifiability and users should be able to adapt the design according to their needs. Here I see potential for optimization in the presented study: the authors only provide Stl-files for 3D printing, so that customization of the design will be exhausting. I find the design very interesting and have ideas about how I could use parts of the design for applications that suit my research (e.g., modifying the workspace beyond well-plates). For this, the community would need the native CAD-files, which is why I ask to make them part of the available source code. I am sure the

authors are aware of this as they have also described possible modifications "LEDs for optogenetic experiments, electromagnets for magnetic cell separation experiments, or anode/cathode pairs for electroporation".

Thank you for the suggestion. We have now included the original source files (.dwg) for all components on our public github repository. We have also updated the manuscript to include the location of this repository.

The authors report predominantly good benchmarks for their system: No cross-contamination and good pipetting performance for high volumes (>25 μL), low cell loss when washing flow-sensitive cells, no differences to cultivation in an ordinary incubator. However, 20% gravimetric error at 10 μL seems to be too much for practical utilization. 10 μL is still a common volume for working in 96-well plates. I am sure that the pipetting performance can be improved here. As I understand it, this error was determined as part of the calibration of the pipette. What is not quite clear to me is how the liquid delivery takes place during calibration (and also during filling of the plate): Was the liquid pipetted into the air, onto the wall, or into a pre-loaded liquid? Was calibration performed in the same way as pipetting into the plates later? Possibly the authors should use a finer pipette tip for this and pipette below the liquid surface. The 23-gauge needle used as pipette is not really the smallest option possible. With direct displacement liquid transfer, there must not be any air cushion in front of or behind the dispensed liquid. How was the pipette tip primed before performing a liquid transfer?

Thank you.

During these calibration procedures liquids were pipetted onto the walls of empty vessels. We have added this information to the materials and methods section. Later pipetting procedures were conducted with liquids pipetted into the bottoms of empty well plates. Pipet tips were primed prior to a liquid transfer by pumping liquids into the pipet tips until a small amount was ejected. We expect most experiments to require higher flow rates and thus optimized PHIL here for larger pipet diameters as the best compromise between cost, speed, accuracy, and ease of use.

Indeed, there are several possible design modifications to increase PHIL's accuracy at smaller volumes. We have added the following text to the discussion regarding how users interested in achieving smaller volume pipetting can do so.

"Improved pumping resolution while pipetting small volumes can be achieved through a variety of methods e.g. by utilizing small diameter hydrophobic tubing as pipets or open-source syringe pumps which include geared stepper motors."

In general: How deep does the pipette tip dip into the reservoir during aspiration and liquid delivery? Is this variable for different applications? Such information should best be explained at the beginning of the result section as part of the system design. Further, how can this be influenced by the user? Is the pipette tip lowered simultaneously during aspiration so that it always aspirates just below the liquid surface to avoid turbulence? Some commercial devices ensure this by using conductive tips. In my experience, this can be solved by the calculated volume + a buffer height (Z-position defined by the filling height of the plate).

Thank you. We have added a brief description of the pipetting process to the results section "Robust media exchange in multi-well plates via automated pipetting" and included a brief discussion of automated pipetting variations that can be programmed with PHIL in the discussion section.

From the Results section:

“To test PHILs’ capacity to change media composition in standard cell culture vessels, we programmed PHIL to position pipets 0.5 mm above 96 well plate bottoms before aspirating 100 μ l PBS and adding 100 μ l 100 μ M FITC at 50 μ L/s and imaged the resulting fluorescence.”

From the Discussion Section:

“Pipetting procedures automated by PHIL are not limited to the simple paradigm utilized here in which the pipet tip is always placed at the well bottom. With minor programming changes, PHIL could also conduct a variety of improved pipetting procedures such as low turbulence pipetting, where the pipet tip is continuously lowered during aspiration and raised during addition so that it is always just below the liquid surface.”

I am not aware of any ISO guideline for the calibration of peristaltic pumps in the specific scope presented here. However, for the applications described here, usually, air displacement and direct displacement pipettes are used, which are regulated according to ISO8655. Authors are encouraged to consult this or another appropriate guideline and clarify the description of the actual pipetting procedure/calibration.

Thank you. We have clarified the calibration process for our peristaltic pumps by adding the following text to the materials and methods section “Pump Calibration.”

“Pumped volume accuracies were determined by pumping 10, 25, 100, and 1000 μ Ls of diH₂O onto the walls of empty 1.5 mL MaxyClear snaplock microcentrifuge tubes (MCT-150-A, Corning Inc.) which were weighed before and after to determine the true pumped volume ($n = 10$ repetitions per condition).”

At the beginning of the result section (design chapter), some other important parameters of the system seem to be missing, which would be important for a technically interested reader: What is the maximum z-stroke of the pipetting arm? The dimensions of the pipette and the PTFE tubing might also be mentioned here. How much μ L volume one micro step of the motors corresponds to would be useful to know. In fact, we were able to achieve excellent pipetting accuracy on a syringe pump in the sub-microliter range by using a NEMA motor equipped with an additional gearbox.

Thank you. We have included further information about the technical specifications of PHIL in results sections “Design of a stage top robot for automated cell culture interventions« and “Robust media exchange in multi-well plates via automated pipetting.”

“Robot arms are raised and lowered up to 45 mm by two stepper motors placed on each side of the manipulator arms and connected to the arm carrying platform via a 3D printed screw drive mechanism (Suppl. Fig. 2).”

“The 23 gauge stainless steel tubes, utilized as pipets, are tightly clustered in a mounting bracket held by the robotic arm. Each pipet is connected to a custom-designed peristaltic pump and media reservoir via 1.07 mm diameter Teflon tubing (Fig. 1c).”

“Based on the dimensions of our peristaltic pump, the theoretical limit of resolution is 0.4 μ L/step.”

A maximum height of 45 mm for target vessels limits the applications of the system but seems ok as a compromise for a short optical path. For flat vessels other than well-plates, however, 45 mm seem rather small. Other users may want to rinse the pipette in external rinsing vials, then you do not have to rinse into unused wells of the well-plate.

Thank you. Indeed, compatibility with live cell microscopy and short optical paths were one critical boundary condition for the design. However, PHIL’s modular 3D printable design enables users to quickly modify characteristics such as chamber geometry to better suit their

needs. We have emphasized this opportunity for customization in the discussion section of our manuscript.

“The incubation chamber can be rapidly modified to include of other devices such as tube holders for tip rinsing or automated sample aliquoting with minimal design changes.”

The cost mentioned in the manuscript (550€) differs from the order list (719€). Where does this difference come from?

Thank you!

A basic PHIL (with 3 pumps and PET tubing) costs ~ 600 USD, while a PHIL with 10 pumps and PTFE tubing costs 800 USD. We have updated the supplementary files and results section to clarify, and changed all costs to USD currency.

“The off the shelf components for a basic PHIL with 3 pumps and polyethylene terephthalate (PET) tubing cost below 600 USD. A PHIL with 10 pumps and Polytetrafluoroethylene (PTFE) tubing and a pre-assembled printed circuit board (PCB) costs up to 800 USD and can be produced with minor soldering steps.”

The order list seems to be incomplete as well, I could not find items like the microcontroller, PCB shield, tubings, connectors, etc. I also could not find the digital PCB layouts online as proclaimed in the article. Additionally, a technical circuit drawing would be useful (preferably created with open-source software).

Thank you for pointing out this oversight! We have updated our order list with these parts and information.

Reviewer #2 (Remarks to the Author):

Thank you for your time and your helpful suggestions!

General information:

In response to our reviewers and to the feedback from users who have already started producing PHILs in other laboratories following our BioRxiv publication, we have made several modifications to the provided online resources. These include:

- Creating a github.com repository containing:
 - Licensing information.
 - All raw files (.dwg) for PHILs design.
 - 3D printable files (.stl) for PHILs production.
 - User interface files (.m) for PHILs user interface.
 - Printed circuit board files for producing PHILs Arduino shield.
- Providing increased detail regarding purchasing PHIL components including a list for purchasing a basic version of PHIL and a list for an advanced version of PHIL.
- Improving available PHIL construction instructions.

We have also updated our website and manuscript to include these improved resources. We will continue to update our website and github repository in response to relevant user feedback (while always also keeping the previous versions available).

The authors present exciting technology in the form of a DIY liquid handling robot that can perform precise and sensitive liquid manipulations on a microscope stage. This development will likely be of interest to many in the biosciences. The paper is well-written and the characterizations seem thorough. I recommend publication after clarification of a few minor items, below. Congratulations to the authors – I hope my students build and use one for themselves.

-The authors mention that PHIL can be used as an environmental chamber on its own with simple connections to preconditioned air, but there is no data to support this. This is not a trivial claim and requires good temperature monitoring possibly with feedback, airtight material and connections, etc. In our experience even expensive stage-top incubators can regularly fail to support cell viability beyond ~24 hrs for one of number of reasons. Can the authors provide data that PHIL can adequately serve as an incubation chamber? If not, then perhaps its use as such should be left as speculation and the text should focus more on its use in controlled environments (eg a large environmental chamber that surrounds the scope and PHIL).

Thank you. The data presented in Figure 4a of this manuscript was produced by culturing murine embryonic stem cells in a PHIL unit maintained at 37 °C using a feedback-controlled heating unit (Life Imaging Services) to heat air in a simple self-made foamed insulation-board box. Humidity and gas composition were maintained by constantly flowing 1 mL/h mixed gas into PHIL through a previously reported 3D printed temperature controlled air humidifier containing roughly 100 mL of diH₂O at 37 °C (<https://bsse.ethz.ch/csd/Hardware/3DPHumidifier.html>). With this setup, we can successfully culture these cells over several days. We have altered the following text of the discussion section and Methods section to clarify this point.

From the discussion Section:

“Within a temperature-controlled enclosure and connected to external gas and humidity sources, PHIL can independently culture sensitive cell types.”

From the Methods Section:

“ESC culture experiments were conducted on environmentally controlled microscopes maintained at 37 °C using a feedback-controlled heating unit (Life Imaging Services) heating foamed insulation-board boxes. Humidity (>98%) and gas composition (5% CO₂, 5% O₂, 90% N₂, PanGas) were maintained by constantly flowing 1 mL/h mixed gas into PHIL through a previously reported 3D printed temperature-controlled air humidifier²³ containing roughly 100 mL of diH₂O at 37 °C (<https://bsse.ethz.ch/csd/Hardware/3DPHumidifier.html>). Relative humidity was recorded using a data logger (Sensirion, SHT31-DIS-B).”

on first read, I thought the 3-10 reusable pipettes were in a multichannel setup that allowed parallel manipulations in separate wells, rather than a bundle that all went into one well. Perhaps the fact that you have up to 10 channels of control in each well could be better emphasized to make this point more clear for the reader. This was also made confusing by varying descriptions (descriptions include: 1-10 reusable pipettes; can dispense through 1 to 9 separate tube channels; pumps can drive flow through 3-10 pipettes). Why the variable descriptions? This could be clarified

Thank you! PHIL’s pipet bundle can contain between 1 and 10 reusable pipets. 1 of these pipets is reserved for liquid aspiration by our GUI, hence up to 9 of the pipets can dispense liquids. Also, we consider 3 pipets to be the minimum required to conduct an interesting experiment: 1 for aspiration, 1 for a negative control liquid, and 1 for a stimulation liquid. To better explain these points, we have added:

Updated figure 1c legend:

“Peristaltic pumps assembled from 3D printed and commercial parts (left) drive flow through 1-10 separate tubes (middle), each ending in separate reusable pipets (right).”

Updated results section:

“PHIL can be configured to dispense or aspirate liquids through 1 to 10 separate tube channels and easily produced reusable pipets.”

-The expt in 3b could be better explained. Is the Venus FACS panel data from a flow cytometer, or from a microscope after sorting? The text states that the distributions between the left and right plots are the same, but they are not (eg “high” overlaps with “mid” on right plot, but not on left).

Thank you. The Venus FACS is directly from a flow cytometer.

The reason for the differences in ‘overlap’ that you mention are the different dynamic ranges of FACS versus immunofluorescence imaging quantifications. These are as expected: We chose this assays because we had previously done the same by manual pipetting in Figure 4 of Ahmed et al., Stem Cell Reports 2020. Here, we now conduct the immunostaining experiment in panel 3b completely by the PHIL robot, and find very comparable data to the previous manually pipetted previous immunostainings (Figure 4: Ahmed et al., Stem Cell Reports 2020). The point we are making here is that the relative brightness of the different populations is following the same trend in both approaches.

We have updated the legend of Fig 3b with the following text to clarify this point.

“The relative brightness of the different populations followed the same trend when observed via FACS or microscopy and was consistent with previously conducted manual immunostainings on the same populations³¹“

-The cell loss results are very impressive! It would be good to have more details of how the washes are done. For example, are these full washes where all of the liquid is removed? Was there just one wash? Does aspiration speed matter? Does positioning of the aspirator matter (eg in the middle of the well vs the wall). Does positioning of dispensing matter? If these parameters haven't been exhaustively tested, at least the authors could provide these details for the data presented.

Thank you. For these experiments each μ -Slide VI 0.4 channel received a single full wash and pipets were placed in the center of each channel's media reservoirs. Aspiration speed was kept the same as pipetting speed for all conditions. Since cell loss is already minimal, we did not optimize other factors such as pipet position further.

We have included the following details in results section “Automated immunostaining pipetting with PHIL.”

“Complete aspiration of formalin solution and one time dispensing of 100 μ L PBS were conducted in the center of media reservoirs on opposing ends of the channels in order to induce unidirectional flow. Liquid aspiration and dispensing were conducted at the same speed.”

-In fig 5D, it would be good to know how many cells were counted for each condition.

Thank you. We have added cell counts per condition to the legend for figure 5D.

Reviewer #3 (Remarks to the Author):

Thank you for your time and your helpful comments!

General information:

In response to our reviewers and to the feedback from users who have already started producing PHILs in other laboratories following our BioRxiv publication, we have made several modifications to the provided online resources. These include:

- Creating a github.com repository containing:
 - Licensing information.
 - All raw files (.dwg) for PHILs design.
 - 3D printable files (.stl) for PHILs production.
 - User interface files (.m) for PHILs user interface.
 - Printed circuit board files for producing PHILs Arduino shield.
- Providing increased detail regarding purchasing PHIL components including a list for purchasing a basic version of PHIL and a list for an advanced version of PHIL.
- Improving available PHIL construction instructions.

We have also updated our website and manuscript to include these improved resources. We will continue to update our website and github repository in response to relevant user feedback (while always also keeping the previous versions available).

The paper presents an open-source liquid handling robot for cell culture. Its main features are low cost, easy to assemble by non-trained persons and a small footprint that allows the robot to be placed in microscopes and other hardware. The robot has been carefully described and characterized and supplementary materials provide detailed assembly instructions and technical details. The robot has also been tested in simple experiments to characterize cross contamination and reliability. Finally, two real experiments have been carried out (automated immunostaining and automated pipetting of long-term live cell cultures). Both experiments have been compared with experiments carried out by humans, achieving similar performance. The paper is clear and well written.

Open-source laboratory equipment have been published several times before, specially using 3D printing techniques. The main motivations are reducing costs, making lab equipment affordable and satisfying special requirements. However, most of the papers lack of real validation tests or the manufacturing requires advance skills. In this regard, this paper provides an outstanding contribution as provides an easy to build liquid handling robot which has been validated thoroughly in complex experiments. Moreover, the versatility of the liquid-handling robot and its low footprint make it suitable to produce new kinds of experiments that could have not been possible before.

In my experience, persons without a technical background have issues assembling these types of systems. Overtightening screws, bad soldering or wiring issues can be difficult to debug. In this paper, they have avoided crimping operations (for connectors) by buying motors with pre-crimped connectors and soldering male headers directly to cables. This causes that sourcing the stepper motors becomes more difficult (most manufactures do not provide connectors in the motors) and requires soldering skills. Soldering can be a difficult task without help or training and a bad soldering technique can produce bad contacts which are difficult to debug as they can produce intermittent behaviors. Thus, I am a bit concern about the quality of robots built by persons without previous training in mechatronics. While the paper states that 5 robots were built by biology PhD students, I would appreciate if the discussion was expanded with the challenges that they found and if they needed help.

Thank you. We agree that solder-free assembly procedures are preferable for inexperienced users and have made significant efforts to avoid them in our design. However, we believe that there has been some confusion regarding connectors for our stepper motors. Male headers are soldered directly to wires for limit switches and not for stepper motors in the provided instructions. We have modified the assembly instructions to clarify this point. Additionally, we have added the following text to the discussion:

“In our hands PHIL could be rapidly constructed by inexperienced users without prior training. This can further be simplified by outsourcing the circuit board assembly to affordable manufacturers (e.g. pcbway.com, jpcb.com, etc.) by simply uploading the provided circuit production files.”

Additionally, it seems that the PCB was already soldered the video that shows the assembly of the robot. Please, specify if this was taken into account in the time required for the assembly and if the biologists also soldered the PCBs.

Thank you. Indeed, in the video PHIL was assembled with a preassembled PCB (since we assume most inexperienced users will use this option). We have added additional clarification regarding build times in our results section and provided a link to the manufacturing service used to outsource circuit assembly. In our test assemblies by inexperienced PhDs found that circuit board assembly consumed approximately 3 h of the total 7 h build time.

Updated results section:

“To confirm the ease of production by inexperienced users, we tested device production with five separate biology PhD students with no engineering experience in our group. Following the provided instructions (Supplementary Materials and Methods), they successfully assembled devices without circuit boards in 2.5 to 4 hours (Supplementary Movie 2) without further assistance. Circuit boards were either self-fabricated in an additional 3 hours, or prefabricated circuit boards were commercially acquired for ~20 USD (https://www.pcbway.com/project/shareproject/ArduinoMega_14Stepper_V3.html).”

Updated materials and methods:

“Custom arduino shields utilized in all experiments were produced in house, while 3 of the PHIL robots constructed by PhD students were assembled by an external manufacturing firm (https://www.pcbway.com/project/shareproject/ArduinoMega_14Stepper_V3.html, ~20 USD each).”

Finally, the stepper drivers have a potentiometer, which controls the amount of current that flows through the motor, that needs to be adjusted. I could not find any instruction about how to adjust it in the assembly instructions.

Thank you, we have now included links to tutorials on potentiometer adjustment in our assembly instructions.

In order to increase the use of the robot, it would be nice if the authors upload the STL files to some online 3D printing services. Thus, persons who want to build the robot would not require access to a 3D printer. The same could apply to the PCB board, which seems the most challenging step for me (e.g. Macrofab, Aisler or Seeed Studio offer manufacturing and assembling of PCBs). These actions could decrease the difficulty to build the robot, increasing the impact of the paper.

Thank you. We did in fact consider uploading our .stl files to manufacturing services to facilitate PHIL production for other users but refrained for two reasons. Firstly, while our desktop 3D printer utilizes fused filament fabrication (FFF) to produce hollow and low weight parts, the manufacturing services currently available generally utilize selective laser melting (SLM) which produces solid and heavy parts. A PHIL robot produced with SLM would require design changes to remain lightweight enough for the microscopy stage-top applications discussed here. Secondly, the suggested manufacturing services generally allow the owner of a design to charge a commission for its production. We felt it would introduce a potential conflict of interest if we were to provide links to services which would then pay us. Also, we wouldn't want to influence a user's choice in manufacturing service. Since we make the .stl files available, users interested in outsourcing the production of a PHIL robot will not find it difficult to find a manufacturing service.

To aid inexperienced users in finding possible services, we have now included the following text in the manuscript:

"Users who wish to outsource the 3D printing of PHIL components can do so by uploading the respective .stl files to a variety of online services (e.g. shapeways.com, hubs.com, i.materialise.com, etc)."

I found strange that the control software of the open-source robot was designed in Matlab, which is a commercial product. This limits the use of the robot to laboratories with a license or they would need to create their own high-level control, which requires specialized knowledge. Have you considered using GNU Octave instead?

Thank you. We chose MATLAB due to its extremely widespread availability in labs and ease of use. It is commonly used in many publications and most students now receive basic training in its use. Licences are free to students and very cheap for academics.

While the paper focuses on cell cultivation, I wonder if it could be applied to other kinds of experiments. I guess that DNA or bacteria experiments are probably out of scope of the machine as avoiding cross contamination is critical in these kinds of experiments.

Thank you! We already included PHIL's use for immunostaining experiments. In addition, PHIL will indeed be useful other applications and have expanded the discussion to reflect that.

From the discussion:

"PHIL will also be useful for many other applications not tested here. For example, the modular design of the pipets and robot arm also allows the inclusion of novel actuators including LEDs for optogenetic experiments, electromagnets for magnetic particle separation experiments, anode/cathode pairs for electroporation, or direct mechanical stimulation of artificial tissues."

Reviewers' Comments:

Reviewer #1:

Remarks to the Author:

The authors are thanked for their thorough revision. I agree with the changes made and recommend the publication of the manuscript in its current form.

Reviewer #2:

Remarks to the Author:

The authors have addressed all of my concerns, which were minor to begin with, and I recommend the paper should be published. Congratulations on excellent work!

Reviewer #3:

Remarks to the Author:

Thanks for the reviewed version of the paper. The repository is indeed a good idea. I think that the article has improved and deserves publication.

Minor comments:

I would explicitly mention that the 5 robots assembled by biology PhD students worked without problems.

Please, state the software and version used for designing the robot in the repository.

Add the native files for the PCB and state the program used to design it. Thus, other people could modify your board. Currently, only the Gerber files are available.

I could not find the link for adjusting the current on the stepper drivers.

REVIEWERS' COMMENTS

Reviewer #1 (Remarks to the Author):

The authors are thanked for their thorough revision. I agree with the changes made and recommend the publication of the manuscript in its current form.

Thank you for your valuable feedback during this process. Your suggestions have helped to improve our manuscript.

Reviewer #2 (Remarks to the Author):

The authors have addressed all of my concerns, which were minor to begin with, and I recommend the paper should be published. Congratulations on excellent work!

Thank you very much! We appreciate your input.

Reviewer #3 (Remarks to the Author):

Thanks for the reviewed version of the paper. The repository is indeed a good idea. I think that the article has improved and deserves publication.

Thank you for your suggestions. They have improved the accessibility of our design.

Minor comments:

- I would explicitly mention that the 5 robots assembled by biology PhD students worked without problems.

Thank you, we have updated the results section "Production and assembly of PHIL robots":

"Following the provided instructions (Supplementary Materials and Methods), they successfully assembled functional devices without circuit boards in 2.5 to 4 hours (Supplementary Movie 2) without further assistance."

- Please, state the software and version used for designing the robot in the repository.

We have updated our repository to reflect the fact that our device was designed in AutoCAD 2018.

- Add the native files for the PCB and state the program used to design it. Thus, other people could modify your board. Currently, only the Gerber files are available.

We have updated the manuscript and repository to include information on the circuit design software used in this work and added the original EAGLE files for our PCB.

From the Materials and Methods section:

"Custom Arduino shields utilized in all experiments were designed in Autodesk EAGLE 2018 (Autodesk Inc.) and produced in house, while 3 of the PHIL robots constructed by PhD students were assembled by an external manufacturing firm (https://www.pcbway.com/project/shareproject/ArduinoMega_14Stepper_V3.html, ~20 USD each)."

- I could not find the link for adjusting the current on the stepper drivers.

Thank you, we have added a link to a current adjustment tutorial to the repository.